# Spatially-Regularized Entropy for Discriminative Token Merging in Fine-Grained Re-Identification

**Shangze Li** [1]  **Yifan Xu** [1]  **Jingmiao Liang** [1]  **Yongfei Zhang** [1][2]  **Yuzhuo Ma** [1]  **Yingbo Qu** [1]

## Abstract

While Vision Transformers (ViTs) offer strong global modeling, their quadratic computational cost limits utility in latency-sensitive applications like person re-identification (ReID). Existing compression strategies, such as token pruning or generic merging, typically rely on coarse-grained criteria tailored for image classification. In fine-grained retrieval, these approaches often discard or smooth out subtle but discriminative local details. To resolve this, we propose SRE-Merge, a training-free framework designed for discriminative token compression. SRE-Merge injects spatial priors into the merging process through three mechanisms: (i) Spatial-Entropy Saliency Assessment (SES-Assess), which quantifies token importance as Spatial-Entropic Mass (SE-Mass) by coupling spatial structure with local attention entropy; (ii) Hybrid Context-Affinity Matching (HCA-Match), which guides precise pair selection by combining feature similarity with mass-derived context; and (iii) Energy-Preserving Weighted Fusion (EPW-Fuse), which incorporates SE-Mass weighting to counteract feature variance reduction. Extensive experiments on standard benchmarks show that SRE-Merge reduces GFLOPs of the base ViT model by about 24% while retaining competitive retrieval accuracy, establishing a superior accuracy-efficiency trade-off.

## 1. Introduction

Person re-identification (ReID) (Ye et al., 2021; Wu et al., 2023a; Yuan et al., 2025c; Liu et al., 2025) has emerged as a key problem in computer vision, aiming to retrieve individuals across multiple non-overlapping cameras. This task is widely applied in public security and intelligent surveillance where real-time response is often required. Recent progress in Vision Transformers (ViTs) has brought significant advances to ReID via strong global modeling (He et al., 2021; Wang et al., 2022a; Zhu et al., 2022; 2024; Li et al., 2022; Yuan et al., 2025b;d). However, ViTs are computationally expensive: multi-head self-attention incurs quadratic complexity with respect to the number of tokens, while feed-forward networks scale linearly with the token count but still account for substantial computation since they are applied to every token with high-dimensional projections. In practice, numerous tokens originate from background regions with minimal discriminative contributions or semantically homogeneous foreground areas, leading to redundant computation and slow inference. These limitations hinder the deployment of ViTs in real-world scenarios.

To reduce this overhead, recent lightweight ReID studies have mainly adopted model pruning techniques (Wu et al., 2023b; Cao et al., 2024; Wang et al., 2024). The key idea is to discard background or redundant tokens that contribute little to discrimination, as seen in RCCReID (Wang et al., 2023) and SUReID (Song & Liu, 2024). Despite their efficiency, pruning permanently removes tokens, disrupting the spatial and contextual continuity essential for fine-grained retrieval. For ReID, identity matching depends not only on global appearance but also on the contextual consistency among local regions. When tokens are discarded based on coarse criteria, the continuity of such context is broken, making it difficult for the model to capture discriminative patterns. Therefore, an ideal strategy should reduce redundancy without sacrificing the contextual integrity of the feature map.

A moderate alternative to pruning is token merging, which compresses the sequence by aggregating similar tokens rather than discarding them (Bolya et al., 2023; Feng & Zhang, 2023; Yoo et al., 2024; Choi et al., 2024; Yuan et al., 2025a). Because merging retains relationships among tokens, it better preserves spatial and contextual information and is, in principle, more compatible with ReID. However, most existing merging strategies are primarily designed for image classification and rely on coarse-grained criteria (e.g., class-token similarity), where preserving category-level semantics is often sufficient. When applied to fine-grained

[1]School of Computer Science and Engineering, Beihang University, Beijing, China [2]State Key Laboratory of Virtual Reality Technology and Systems, Beihang University, Beijing, China. Correspondence to: Yongfei Zhang <yfzhang@buaa.edu.cn>.

*Proceedings of the 43rd International Conference on Machine Learning*, Seoul, South Korea. PMLR 306, 2026. Copyright 2026 by the author(s).

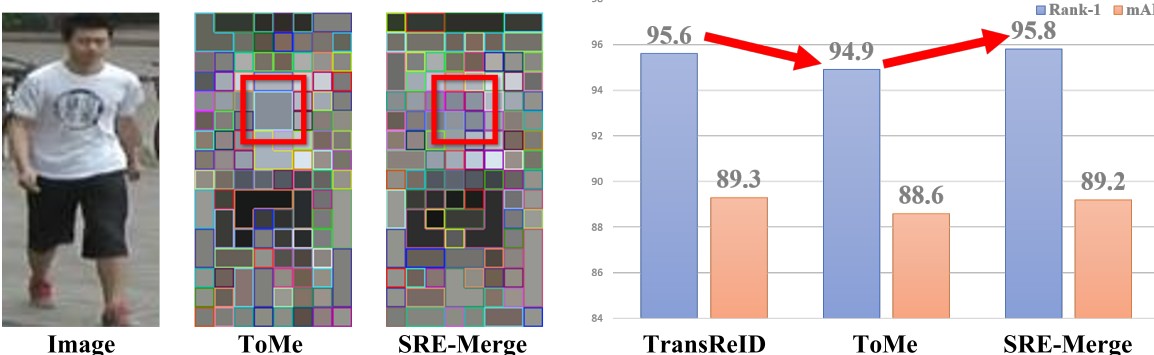

*Figure 1.* Directly applying generic token merging techniques results in an apparent performance drop for ReID, due to their inability to retain critical retrieval cues, as highlighted by the red boxes. In contrast, SRE-Merge effectively preserves this crucial information, thereby maintaining high accuracy. Note that SRE-Merge reduces GFLOPs to 15.53, which is **23.9%** lower than TransReID (20.41).

retrieval, however, such approaches fail to distinguish subtle details: First, regarding *token assessment*, current methods ignore spatial structure and varying information density, leading to the underestimation of critical local cues. Second, regarding *token matching and fusion*, operations driven purely by feature similarity can mistakenly merge crucial tokens (e.g., face, logos) with background or cause feature variance reduction during averaging, leading to representation collapse. As Figure 1 implies, operations harmless for classification may remove the very signals ReID requires.

To resolve this, we propose **SRE-Merge**, a training-free framework designed for discriminative token compression. SRE-Merge injects spatial priors into the merging pipeline to preserve local details through three mechanisms. First, **Spatial-Entropy Saliency Assessment (SES-Assess)** is proposed to overcome assessment bias, which quantifies token importance as **Spatial-Entropic Mass (SE-Mass)** by coupling spatial structure with local attention entropy. Second, we design **Hybrid Context-Affinity Matching (HCA-Match)**, which guides precise pair selection by combining feature similarity with mass-derived context. Finally, **Energy-Preserving Weighted Fusion (EPW-Fuse)** incorporates SE-Mass weighting and variance-preserving constraints to counteract feature variance reduction inherent in averaging operations.

Our main contributions are summarized as follows:

- We introduce SRE-Merge, a novel training-free token merging framework for ViT-based ReID, which performs spatially-regularized entropy-guided merging to reduce token redundancy while preserving identity-discriminative details.

- We design SES-Assess, a strategy that quantifies token importance as Spatial-Entropic Mass (SE-Mass) by integrating spatial priors with local entropy, ensuring crucial tokens are accurately evaluated.

- We develop HCA-Match and EPW-Fuse, which leverage mass-derived context and variance-preserving constraints to mitigate erroneous fusion of discriminative tokens and prevent representation collapse.

- Extensive experiments validate that SRE-Merge achieves a significant reduction in computational cost while maintaining accuracy comparable to state-of-the-art methods, yielding a superior accuracy-efficiency trade-off.

## 2. Related Work

**Token Pruning.** The token pruning strategy compresses the input sequence by removing low-score tokens. For example, DynamicViT (Rao et al., 2021) introduces a lightweight prediction module to estimate token importance scores and dynamically prune low-score tokens at each layer. EViT (Liang et al., 2022) uses a hierarchical approach, classifying tokens into valid and invalid categories based on attention scores, retaining the valid tokens and fusing the invalid ones to reduce redundancy. SPViT (Kong et al., 2022) adopts a stage-wise, layer-by-layer training strategy to progressively prune tokens and optimize model performance.

**Token Merging.** The token merging strategy achieves sequence compression by combining multiple tokens into a single new token. Unlike the pruning strategy, the merging strategy retains more information, thus better maintaining recognition accuracy. For instance, ToMe (Bolya et al., 2023) and Token Pooling (Marin et al., 2023) merge tokens based on semantic similarity, effectively reducing redundant information. BAT (Long et al., 2023) first prioritizes tokens based on their importance, and then selectively retains tokens according to the principle of diversity. These approaches enhance overall efficiency while ensuring the model retains key information.

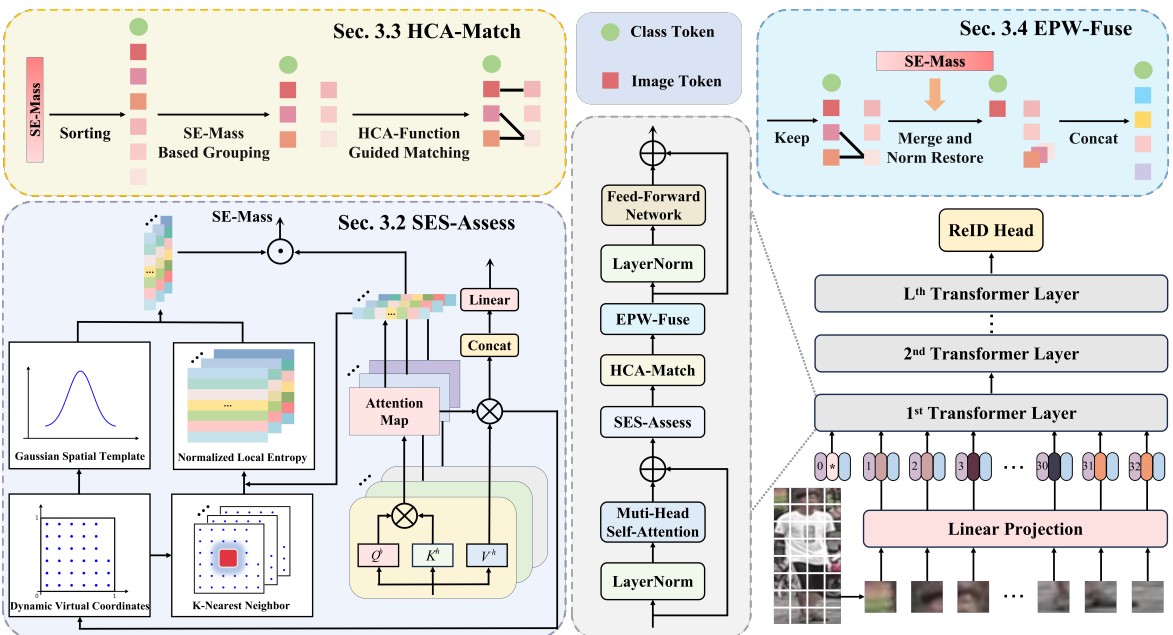

*Figure 2.* The overall architecture of the proposed **SRE-Merge** framework. It operates in a training-free manner through three sequential stages: (1) **SES-Assess** first evaluates the information density of each token by coupling spatial priors with local attention entropy, generating **Spatial-Entropic Mass (SE-Mass)** scores; (2) **HCA-Match** then utilizes these mass scores to derive contextual influence, which is combined with feature similarity to guide the precise matching of redundant tokens; (3) **EPW-Fuse** finally aggregates the matched pairs using mass-weighted fusion and performs scale restoration to counteract feature variance reduction, ensuring the preservation of discriminative representations.

**Lightweight Person Re-Identification.** Token compression methods in lightweight ReID can be divided into two categories. The first category of methods relies on human structural priors. For instance, RCCReID (Wang et al., 2023) utilizes an edge detection network (Xie & Tu, 2015) to extract image edge maps, and filters background regions by selecting valid tokens, while maintaining recognition accuracy and reducing computational cost. The other category of methods requires no external priors. PAPReID (Ndayishimiye et al., 2025) introduces a dynamic token selection module, which evaluates importance scores to filter tokens and retains only key tokens for computation. SUReID (Song & Liu, 2024) proposes a hierarchical token sparsification strategy that generates token importance scores, makes differentiable binarization decisions, and dynamically updates masks to progressively eliminate redundant tokens. These methods reduce computational overhead while maintaining accuracy, promoting the development of lightweight ReID.

## 3. Methodology

In this section, we formally present our **SRE-Merge** framework, as illustrated in Figure 2. We begin by briefly revisiting TransReID (He et al., 2021), which provides the infrastructure of our approach (Section 3.1). Then, we elaborate on the **Spatial-Entropy Saliency Assessment (SES-Assess)** strategy (Section 3.2), which quantifies token importance

as Spatial-Entropic Mass (SE-Mass). Next, we introduce the **Hybrid Context-Affinity Matching (HCA-Match)** scheme (Section 3.3), which guides precise pair selection by combining feature similarity with mass-derived context. Finally, we propose the **Energy-Preserving Weighted Fusion (EPW-Fuse)** paradigm (Section 3.4), which counteracts feature variance reduction.

### 3.1. Revisit TransReID

Given an image $x \in \mathbb{R}^{H \times W \times C}$, where $H$, $W$, and $C$ represent its height, width, and number of channels, respectively, it is split into $N$ fixed-sized patches through a sliding window. A learnable class token [cls] of dimension $D$, denoted as $z_{\text{cls}} \in \mathbb{R}^D$, is added at the beginning of the sequence. Position embeddings and side information embeddings are then incorporated to form the token sequence $Z_0 \in \mathbb{R}^{(N+1) \times D}$, which is passed through $L$ transformer layers.

Within each transformer layer, the computational process can be formulated as follows:

$$Z'_l = \text{MHSA}(\text{LN}(Z_{l-1})) + Z_{l-1}, \quad l \in [1, L], \quad (1)$$

$$Z_l = \text{FFN}(\text{LN}(Z'_l)) + Z'_l, \quad l \in [1, L], \quad (2)$$

where $\text{MHSA}(\cdot)$ represents multi-head self-attention (Vaswani et al., 2017), $\text{FFN}(\cdot)$ denotes feed-forward network, and LN is layer normalization. Let $Q, K, V \in$

$\mathbb{R}^{(N+1)\times D}$ denote the query, key, and value matrices, MHSA performs parallel attention computation on the input:

$$\text{Attention}(Q, K, V) = A \times V, \quad A = \text{Softmax}\left(\frac{QK^T}{\sqrt{D_h}}\right),$$
$$(3)$$

where $D_h = D/H$ denotes the output dimension of a single head among $H$ attention heads. Similarly, the class-token attention scores are given by:

$$A(z_{\text{cls}}, :) = \text{Softmax}\left(\frac{Q_{\text{cls}}K^T}{\sqrt{D_h}}\right). \qquad (4)$$

Current token assessment approaches typically average outputs across all attention heads, while neglecting spatial and contextual relations between tokens. This often leads to the underestimation of discriminative tokens in critical local regions, thereby losing crucial retrieval cues.

### 3.2. Spatial-Entropy Saliency Assessment

To mitigate information loss caused by naive head averaging, we propose **SES-Assess**, which computes an entropy measure for each spatial location and uses local neighborhood entropy to differentiate the saliency of each attention head dynamically. Attention heads that exhibit lower entropy, which indicate more concentrated distributions (higher confidence), receive higher weights. This approach relies only on MHSA outputs and requires no extra supervision, enabling a **training-free** assessment mechanism (more details are in the **Appendix**).

TransReID uses the sliding window to generate patches, and subsequent merging may change the number and arrangement of tokens across layers. Traditional approaches based on fixed dimensions or positions are rendered inapplicable, as they cannot predetermine the variable token configurations (quantity and sequence) at each layer. To establish spatial location relationships, we introduce *Dynamic Virtual Coordinates* (DVC) that map an arbitrary number of tokens onto a virtual grid. Given $N$ image tokens at the current layer (excluding the class token), we arrange them sequentially into a virtual $H_v \times W_v$ grid with $H_v = \lceil\sqrt{N}\rceil$, $W_v = \lceil N/H_v \rceil$, and index tokens by $i \in \{0, \dots, N-1\}$. The normalized coordinates of token $i$ are $x_i = \frac{\lfloor i/W_v \rfloor + 0.5}{H_v}$, $y_i = \frac{i\%W_v + 0.5}{W_v}$, so that $(x_i, y_i) \in (0, 1)^2$ and token layout reflects the virtual spatial arrangement. After each merging operation, DVC is recomputed for the current token sequence, where each fused token is treated as a single token in the next layer; thus, DVC provides a layer-adaptive coarse spatial layout rather than fixed pixel-level coordinates.

In ReID datasets, images are constructed by cropping individual pedestrian silhouettes based on object detectors or manual annotations. Due to the high precision of modern detectors, foreground pedestrians critical for identity

recognition are typically centered, meaning central regions contain more salient information. To encode this prior, we define *Adaptive Gaussian Spatial Template* (AdaptGST) that emphasizes central regions. For a layer with $N$ tokens, we employ the Gaussian scale $\sigma = \alpha\sqrt{N}$ (with $\alpha$ conventionally set to 0.1), and the spatial prior for token $i$ is

$$\mathbf{W}_{\text{space}}(x_i, y_i) = \exp\left(-\frac{(x_i - 0.5)^2 + (y_i - 0.5)^2}{2\sigma^2}\right).$$
$$(5)$$

This template dynamically and adaptively adjusts the central prior intensity according to the token density at the current layer, enabling adaptation to varying spatial distributions of salient information.

To eliminate dependency on fixed grids and flexibly perceive spatial and contextual relations, we form for each token $z_i$ a neighborhood $\mathcal{N}(i)$ consisting of its $k$ nearest tokens under Euclidean distance in the virtual coordinates $(x_i, y_i)$. Let $\mathbf{A}^{(h)}(z_{\text{cls}}, z_j)$ denote the attention from the class token $z_{\text{cls}}$ to neighboring token $z_j$ in head $h$. Since attention weights are originally normalized over the entire token set, we first re-normalize them within the local neighborhood $\mathcal{N}(i)$ to obtain a valid local probability distribution:

$$p_{i,j}^{(h)} = \frac{\mathbf{A}^{(h)}(z_{\text{cls}}, z_j)}{\sum_{u \in \mathcal{N}(i)} \mathbf{A}^{(h)}(z_{\text{cls}}, z_u)}, \quad j \in \mathcal{N}(i). \quad (6)$$

Then, the local neighborhood entropy for head $h$ at token $z_i$ is computed as

$$\mathcal{H}^{(h)}(z_i) = -\frac{1}{\log|\mathcal{N}(i)|} \sum_{j \in \mathcal{N}(i)} p_{i,j}^{(h)} \log p_{i,j}^{(h)}. \quad (7)$$

When $|\mathcal{N}(i)| = 1$, the local distribution degenerates to a deterministic one, and we set $\mathcal{H}^{(h)}(z_i) = 0$ to avoid the undefined normalization factor $\log|\mathcal{N}(i)|$. This normalization ensures that $\mathcal{H}^{(h)}(z_i)$ is computed on a valid probability distribution and remains comparable across neighborhoods with different sizes. A lower entropy indicates a more concentrated and confident attention distribution of head $h$ on the corresponding neighborhood. Subsequently, we derive the entropy-aware attention weight of head $h$ for token $z_i$ through a normalized exponential function:

$$\mathcal{R}^{(h)}(z_i) = \frac{\exp\left(-\mathcal{H}^{(h)}(z_i)\right)}{\sum_{h'=1}^{H} \exp\left(-\mathcal{H}^{(h')}(z_i)\right)}. \quad (8)$$

This formulation dynamically allocates the contribution of each attention head according to its local neighborhood entropy. Compared to conventional averaging approaches, our method amplifies contributions from heads with more concentrated and confident local responses, thereby achieving enhanced focus on crucial details.

Ultimately, the importance of token $z_i$ is defined as the entropy-weighted aggregation of class-token attention

across all heads, further modulated by the spatial prior. We define this quantity as the **Spatial-Entropic Mass (SE-Mass)**, denoted as $\mathcal{S}(z_i)$ ($i \in [1, N]$), which serves as the fundamental physical quantity guiding the subsequent matching and fusion processes:

$$\mathcal{S}(z_i) = \mathbf{W}_{\text{space}}(x_i, y_i) \sum_{h=1}^{H} \mathcal{R}^{(h)}(z_i) \cdot \mathbf{A}^{(h)}(z_{\text{cls}}, z_i). \quad (9)$$

Here, $\mathbf{W}_{\text{space}}(x_i, y_i)$ injects the pedestrian-centered spatial prior at the token-saliency level, while $\mathcal{R}^{(h)}(z_i)$ adaptively emphasizes attention heads with more confident local responses for each token.

### 3.3. Hybrid Context-Affinity Matching

**SE-Mass-Based Grouping.** After computing the SE-Mass for each image token, we keep the class token $z_{\text{cls}}$ separate and consider the set of image tokens indexed by $\mathcal{I} = \{1, \ldots, N\}$. Let $\mathcal{S} = \{\mathcal{S}(z_i)\}_{i \in \mathcal{I}}$ be the SE-Mass values for these tokens, we form a descending permutation Index $= (p_1, \ldots, p_N)$ by sorting $\mathcal{S}$ in decreasing order, i.e., $\mathcal{S}(z_{p_1}) \geq \mathcal{S}(z_{p_2}) \geq \cdots \geq \mathcal{S}(z_{p_N})$. Then we split this ordered list into two subsequences by alternating positions: $\text{Index}_A = \{0, p_1, p_3, p_5, \ldots\}$, $\text{Index}_B = \{p_2, p_4, p_6, \ldots\}$, where the class-token index 0 is explicitly inserted into $\text{Index}_A$ to help preserve core features during matching. Based on this partitioning, we extract the corresponding tokens from $Z \in \mathbb{R}^{(N+1) \times D}$ and form two subsequences $Z_A = Z[\text{Index}_A]$ and $Z_B = Z[\text{Index}_B]$. The traditional fixed odd-even grouping may lead to semantically similar or redundant tokens being clustered within the same group, thereby hindering cross-group pairing (as matching occurs exclusively between groups). Our SE-Mass metric reflects task-specific priority of tokens. This priority-ordered grouping strategy effectively disrupts fixed token distributions and enhances semantically correlated merging.

**Hybrid Context-Affinity Function.** Given $Z \in \mathbb{R}^{(N+1) \times D}$, the goal at each layer is to merge $r$ pairs of redundant tokens and produce a compressed sequence $\hat{Z} \in \mathbb{R}^{(N+1-r) \times D}$. To guide matching while protecting identity-relevant crucial tokens, we define a **Hybrid Context-Affinity Function (HCA-Function)** that combines feature similarity with mass-derived context to evaluate inter-group token relationships.

Since the $K$ matrix has distilled critical information of each token through self-attention, we use the $K$ vectors (rows of the $K$ matrix) as compact semantic descriptors for feature similarity. For tokens $z_i$ and $z_j$ with $K$ vectors $k_i, k_j \in \mathbb{R}^D$, the cosine-based similarity is defined as:

$$F_{\text{sim}}(k_i, k_j) = \frac{1}{2} \left( \frac{k_i \cdot k_j^T}{\| k_i \|_2 \cdot \| k_j \|_2} + 1 \right), \quad (10)$$

where $\| \cdot \|_2$ represents the L2-norm. The final results are normalized and shifted to the [0,1] interval.

However, relying solely on inter-token similarity as the merging criterion entails potential risks. Semantically similar tokens in feature space are not necessarily redundant, particularly when they are located in critical local regions essential for identity modeling. Erroneous merging of such tokens may lead to irreversible loss of discriminative details, ultimately degrading model performance. Therefore, we design the *Contextual Influence* metric as an auxiliary matching criterion to evaluate each token's contribution to the identity representation. The aforementioned SE-Mass is leveraged to formulate the contextual influence score $F_{\text{res}}$. Specifically, a higher SE-Mass signifies that the token possesses stronger semantics for identity discrimination, thus should be preserved from merging so as to mitigate the risk of losing crucial details. Conversely, a token with a lower SE-Mass indicates diminished global influence and is likely a semantically redundant token. The contextual influence score between token pairs is expressed as follows:

$$F_{\text{res}}(z_i, z_j) = \hat{\mathcal{S}}(z_i) \cdot \hat{\mathcal{S}}(z_j), \quad (11)$$

$$\hat{\mathcal{S}}(z_i) = \frac{1/\mathcal{S}(z_i)}{\max_k (1/\mathcal{S}(z_k))}, \quad k \in [1, N]. \quad (12)$$

The rationale behind this product form is to enforce a "criticality preservation" constraint: if even one token in a pair is critical (high SE-Mass $\to$ low $\hat{\mathcal{S}}$), the product $F_{\text{res}}$ remains low. This effectively reduces the matching priority of pairs containing critical tokens, avoiding the misintegration of discriminative cues into redundant ones. The final HCA-Function value is denoted as the product of the similarity and contextual influence with tunable coefficients:

$$F(z_i, z_j) = (F_{\text{sim}}(k_i, k_j))^{\delta_1} \cdot (F_{\text{res}}(z_i, z_j))^{\delta_2}, \quad (13)$$

where $\delta_1, \delta_2 > 0$ control the relative emphasis on semantic similarity and contextual influence. A higher HCA-Function value indicates a greater probability of token fusion between the corresponding pair.

**Bipartite Soft Matching with HCA-Function.** We perform bipartite soft matching (Bolya et al., 2023) between tokens in $Z_A$ and $Z_B$ using the HCA-Function. First, we compute the matrix of correlation scores $F(z_i, z_j)$ for all $z_i \in Z_A$ and $z_j \in Z_B$. Then, we identify the candidate pairs by selecting the highest-scoring matches. Finally, all candidate pairs are sorted in descending order of scores, and the top-$r$ highest-scoring token pairs are selected as the ultimate fusion targets, which are then passed to the fusion module (EPW-Fuse). For more details of the algorithm, please refer to the **Appendix**.

*Table 1.* Comparison of GFLOPs, inference time (s), throughput (img/s), Rank-1 (%), and mAP (%) with state-of-the-art token compression methods on MSMT17 and Market-1501.

| Methods | GFLOPs ↓ | Inference Time ↓ | Throughput ↑ | MSMT17 | | Market-1501 | |
|---|---|---|---|---|---|---|---|
| | | | | Rank-1 ↑ | mAP ↑ | Rank-1 ↑ | mAP ↑ |
| Specific Compression Methods for ReID | | | | | | | |
| RCCReID (Wang et al., 2023) | 17.18 | - | - | 82.2 | 63.0 | 94.9 | 88.0 |
| SUReID (Song & Liu, 2024) | - | - | - | - | - | 94.5 | 87.2 |
| PAPReID (Ndayishimiye et al., 2025) | 16.86 | - | - | - | - | 79.1 | 77.3 |
| Generic Compression Methods for ViTs | | | | | | | |
| DPC-KNN (Du et al., 2016) | 17.24 | 54.79 | 351.89 | 84.5 | 65.7 | 94.7 | 87.9 |
| DynamicViT (Rao et al., 2021) | 19.30 | 58.31 | 330.68 | 84.5 | 66.6 | 94.7 | 87.5 |
| EViT (Liang et al., 2022) | 17.50 | 54.27 | 355.27 | 84.9 | 67.0 | 94.8 | 88.0 |
| ATS (Fayyaz et al., 2022) | 17.39 | 55.34 | 348.43 | 81.2 | 58.8 | 93.4 | 84.7 |
| ToMe (Bolya et al., 2023) | **15.53** | **45.99** | **419.25** | 85.0 | 67.1 | 94.9 | 88.6 |
| TransReID (He et al., 2021) | 20.41 | 57.43 | 335.73 | **85.3** | **68.1** | 95.6 | **89.3** |
| **SRE-Merge (Ours)** | **15.53** ↓²³·⁹% | 46.28 ↓¹⁹·⁴% | 416.62 ↑²⁴·¹% | **85.3** | 67.8 | **95.8** ↑⁰·²% | 89.2 |

## 3.4. Energy-Preserving Weighted Fusion

Traditional token-size-based fusion commonly uses weighted averaging to obtain a fused token. Such averaging, however, ignores differences in token importance: semantically important tokens can be overly smoothed by redundant ones, degrading feature discriminability.

To this end, we introduce **Energy-Preserving Weighted Fusion (EPW-Fuse)**, which incorporates SE-Mass into the weighting strategies to mitigate the shortcomings of token-size-based methods and counteract feature variance reduction. For $T$ tokens to be merged, the fused feature representation is defined as:

$$z_{\text{avg}} = \sum_{i=1}^{T} w_i z_i, \quad w_i = \frac{\mathcal{S}(z_i)}{\sum_{k=1}^{T} \mathcal{S}(z_k)}, \quad \sum_{i=1}^{T} w_i = 1. \quad (14)$$

The weighted summation is essentially a linear combination. According to the norm inequality $\|z_{\text{avg}}\| = \left\| \sum_{i=1}^{T} w_i z_i \right\| \leq \max \|z_i\| \, (i \in [1, T])$, the fused norm is inevitably bounded by the maximum norm of the original tokens. From the perspective of representation learning, the L2-norm of a feature vector correlates with its variance or "energy". A decrease in norm implies a contraction of features towards the origin in the hyperspace, which can damage the decision boundaries in softmax-based classification (feature variance reduction). Notably, as the network depth increases, this norm diminishment becomes more pronounced (more analysis can be found in the **Appendix**). Therefore, we employ max-norm linear interpolation for token merging. Building upon SE-Mass weighted fusion, we compute the L2-norm of the fused feature and perform normalization. By selecting the maximum L2-norm from the original features of each token pair, we apply scaling to the normalized fused feature, thereby restoring the scale

information of the feature representation:

$$\hat{z}_{\text{avg}} = \frac{z_{\text{avg}}}{\|z_{\text{avg}}\|_2} \times \max_{1 \leq i \leq T} \|z_i\|_2. \quad (15)$$

This operation acts as a variance-preserving constraint, ensuring that the compressed tokens remain distributed on the high-energy surface of the feature hyperspace, thereby maintaining discriminability.

# 4. Experiments

## 4.1. Datasets and Evaluation Metrics

**Datasets.** We evaluate our method on three public ReID benchmarks: Market-1501 (Zheng et al., 2015) and MSMT17 (Wei et al., 2018), and Occluded-Duke (Miao et al., 2019). The Market-1501 contains 32,668 images of 1,501 identities acquired from 6 surveillance cameras, while the MSMT17 consists of 126,441 images of 4,101 identities captured by 15 cameras. The Occluded-Duke includes 1,812 identities, particularly designed to evaluate model robustness in complex real-world scenarios with heavy occlusion challenges. To further validate the generalization ability of our proposed method on the image classification task, we also conduct comparative experiments between SRE-Merge and token compression methods on the ImageNet-1k (Deng et al., 2009) dataset.

**Evaluation Metrics.** We employ Giga Floating-point Operations (GFLOPs) as the metric to quantify the computational complexity of the model. Besides, we also employ inference time and throughput to evaluate the computational efficiency, which refer to the time required for the model to complete inference on a dataset and the number of images processed per second respectively. Furthermore, we utilize the cumulative matching characteristic (CMC) curve

*Table 2.* Comparison of GFLOPs and Rank-1 (%) with ToMe on Market-1501 (MK), MSMT17 (MS), and Occluded-Duke (OD). SRE-Merge merges $r$=17 tokens per layer, while ToMe adopts different merging numbers $r$.

| *(a)* ToMe $r$=17 | | | | | *(b)* ToMe $r$=11 | | | | | *(c)* ToMe $r$=6 | | | | |
|---|---|---|---|---|---|---|---|---|---|---|---|---|---|---|
| Methods | GFLOPs | MK | MS | OD | Method | GFLOPs | MK | MS | OD | Method | GFLOPs | MK | MS | OD |
| ToMe | **10.31** | 94.4 | 83.5 | 64.2 | ToMe | 13.78 | 94.9 | 84.5 | 65.6 | ToMe | 16.69 | 95.1 | **84.9** | 66.5 |
| SRE-Merge | **10.31** | **95.2** | **84.8** | **67.1** | SRE-Merge | **10.31** | **95.2** | **84.8** | **67.1** | SRE-Merge | **10.31** | **95.2** | 84.8 | **67.1** |
| *Comparison* | ↓0% | ↑0.8% | ↑1.3% | ↑2.9% | *Comparison* | ↓25.2% | ↑0.3% | ↑0.3% | ↑1.5% | *Comparison* | ↓38.2% | ↑0.1% | ↓0.1% | ↑0.6% |

*Table 3.* Comparison with recent VLM/MLLM-oriented training-free visual token reduction methods on ReID benchmarks. Results are reported as Rank-1 / mAP (%).

| Methods | Venue | GFLOPs ↓ | Market-1501 | MSMT17 | Occluded-Duke |
|---|---|---|---|---|---|
| VisionZip (Yang et al., 2025) | CVPR 2025 | 18.75 | 93.0 / 81.8 | 82.2 / 60.6 | 58.9 / 44.9 |
| DivPrune (Alvar et al., 2025) | CVPR 2025 | 17.84 | 94.5 / 88.5 | 84.8 / 67.2 | 66.2 / 58.4 |
| VisionTrim (Yu et al., 2026) | ICLR 2026 | 18.79 | 93.0 / 81.2 | 81.7 / 60.6 | 52.4 / 39.8 |
| ApET (Ma et al., 2026) | CVPR 2026 | 18.79 | 94.3 / 86.4 | 83.4 / 62.7 | 60.3 / 51.1 |
| OTPrune (Chen et al., 2026) | CVPR 2026 | 18.79 | 93.0 / 81.0 | 81.9 / 60.4 | 56.3 / 42.7 |
| **SRE-Merge (Ours)** | - | **15.53** | **95.8 / 89.2** | **85.3 / 67.8** | **67.5 / 59.4** |

and mean average precision (mAP) as metrics to evaluate the retrieval accuracy. Rank-$k$ in the CMC curve measures the probability of finding a correct match within the top-$k$ results, while mAP evaluates average retrieval performance across all queries.

### 4.2. Implementation Details

The proposed SRE-Merge is implemented in PyTorch following TransReID, with all experiments conducted on a single A100 GPU. We employ TransReID as our backbone network, where the image encoder is initialized with ViT-B/16 on ReID datasets. On ImageNet-1k, we adopt three different backbone networks for fair comparison with other methods, namely DeiT-T, DeiT-S, and DeiT-B. All images are resized to 256×128. The batch size is set to 64, with each mini-batch containing 16 identities and 4 images per identity. We select $k$=9 nearest tokens as the neighborhood. Unless otherwise specified, our token merging is applied to layers 1 to 10 of the model, with the number of merged tokens per layer set to $r$=8. Please refer to the **Appendix** for the determination of the parameters $k$ and $r$.

### 4.3. Comparison with State-of-the-Art Methods

To verify the effectiveness of the proposed method, we first conduct experiments on computational complexity and retrieval accuracy using the MSMT17 and Market-1501 datasets. We comprehensively compare SRE-Merge with previous state-of-the-art token compression methods, and all results are reported without re-ranking. For fair comparison, we adopt the same settings as ToMe, i.e., merging $r$=8 tokens per layer, resulting in the same GFLOPs. As shown in Table 1, SRE-Merge's inference efficiency is slightly lower than that of ToMe. This is because ToMe does not evaluate

the semantic importance of tokens and instead directly performs grouping and merging operations. In contrast, we first employ SES-Assess to ensure crucial tokens are retained, thus outperforming ToMe significantly across all four accuracy metrics on both datasets, yet the increased inference overhead is almost negligible. Compared to TransReID, SRE-Merge reduces GFLOPs by 23.9%, shortens inference time by 19.4%, and increases throughput by 24.1%, while achieving comparable accuracy. This indicates that SRE-Merge establishes a favorable balance between recognition accuracy and computational efficiency for lightweight ReID.

For an intuitive and comprehensive comparison with ToMe, we fix the merging number of SRE-Merge as $r$=17 and evaluate ToMe under different merging numbers across three representative ReID benchmarks. As illustrated in Table 2, SRE-Merge reduces the computational cost to 10.31 GFLOPs, which is nearly half of the original TransReID, while still achieving strong Rank-1 on Market-1501, MSMT17, and Occluded-Duke. (a) When ToMe also adopts $r$=17, both methods have the same computational cost, but SRE-Merge consistently achieves higher Rank-1, improving ToMe by 0.8%, 1.3%, and 2.9% on Market-1501, MSMT17, and Occluded-Duke, respectively. This indicates that under the same compression ratio, the proposed ReID-oriented merging strategy preserves identity-discriminative cues more effectively than generic token merging. (b) When ToMe uses a smaller merging number, i.e., $r$=11, SRE-Merge requires only 74.8% of ToMe's GFLOPs and still obtains higher Rank-1 on all three datasets, with improvements of 0.3%, 0.3%, and 1.5%, respectively. This shows that SRE-Merge can achieve better accuracy even under a much lower computational budget. (c) When ToMe further reduces the compression strength to $r$=6 to pursue higher accuracy, SRE-Merge still requires only 61.8% of ToMe's

*Table 4.* Comparison of Rank-1 (%) and mAP (%) with state-of-the-art ReID methods on Market-1501 and MSMT17. **Note that SRE-Merge reduces approximately 24% of GFLOPs, while other methods *DO NOT* perform compression here.**

| Backbone | Methods | Market-1501 | | MSMT17 | |
|---|---|---|---|---|---|
| | | Rank-1 ↑ | mAP ↑ | Rank-1 ↑ | mAP ↑ |
| CNN | AGW (Ye et al., 2021) | 95.1 | 87.8 | 68.3 | 49.3 |
| | DRL-Net (Jia et al., 2022) | 94.7 | 86.9 | 78.4 | 55.3 |
| | MSINet (Gu et al., 2023) | 95.3 | 89.6 | 81.0 | 59.6 |
| | HashReID (Nikhal et al., 2024) | 94.2 | 84.9 | 76.8 | 51.4 |
| | DCR (Yang & Xu, 2025) | 95.3 | 88.3 | 81.0 | 56.4 |
| ViT | PFD (Wang et al., 2022a) | 95.5 | 89.7 | - | - |
| | DCAL (Zhu et al., 2022) | 94.7 | 87.5 | 83.1 | 64.0 |
| | AAformer (Zhu et al., 2024) | 95.4 | 87.7 | 83.6 | 63.2 |
| | ProFD (Cui et al., 2024) | 95.1 | 90.0 | - | - |
| | MV-3DSReID (Yu et al., 2024) | 95.7 | **90.2** | - | - |
| | DPEFormer (Zhang et al., 2025) | 95.4 | 88.1 | - | - |
| | TransReID (He et al., 2021) | 95.6 | 89.3 | **85.3** | **68.1** |
| | **SRE-Merge (Ours)** | **95.8** | 89.2 | **85.3** | 67.8 |

*Table 5.* Comparison of Rank-1 (%) and mAP (%) with state-of-the-art methods on Occluded-Duke. **Note that SRE-Merge reduces approximately 24% of GFLOPs, while other methods *DO NOT* perform compression here.**

| Methods | Rank-1 | mAP |
|---|---|---|
| FED (Wang et al., 2022b) | 68.1 | 56.4 |
| RTGAT (Huang et al., 2023) | 61.0 | 50.1 |
| PRE (Yan et al., 2023) | **68.3** | 55.2 |
| ETND (Dong et al., 2024) | 68.1 | 57.6 |
| MTIPE (Yuan et al., 2025e) | 66.4 | 57.8 |
| **SRE-Merge (Ours)** | 67.5 | **59.4** |

*Table 6.* Comparison of Rank-1 (%) with state-of-the-art token compression methods using multiple backbones on ImageNet-1k.

| Methods | DeiT-S | DeiT-T | DeiT-B |
|---|---|---|---|
| DPC-KNN (Du et al., 2016) | 78.85 | 71.10 | 79.06 |
| DynamicViT (Rao et al., 2021) | 79.17 | 67.40 | 80.68 |
| EViT (Liang et al., 2022) | 79.30 | 71.06 | 80.99 |
| ATS (Fayyaz et al., 2022) | 79.09 | 70.77 | 80.78 |
| ToMe (Bolya et al., 2023) | 79.39 | 71.10 | 81.05 |
| **SRE-Merge (Ours)** | **79.41** | **71.43** | **81.32** |

GFLOPs. Despite this substantial reduction in computation, SRE-Merge achieves slightly higher Rank-1 on Market-1501 and Occluded-Duke, and remains almost on par with ToMe on MSMT17 with only a marginal 0.1% difference. These results demonstrate that SRE-Merge achieves a more favorable accuracy-efficiency trade-off than ToMe across multiple ReID scenarios. The superiority mainly comes from the ReID-specific designs of SES-Assess, HCA-Match, and EPW-Fuse, which jointly avoid over-merging discriminative pedestrian tokens and preserve fine-grained identity information during token compression.

To further evaluate the practical effectiveness of SRE-Merge against recent advances in training-free token reduction, we compare it with five representative VLM/MLLM-oriented visual token pruning or merging methods. For a fair comparison, we port their core token reduction strategies into the same ViT-B-based TransReID backbone and evaluate all methods without additional training. Since SRE-Merge adopts the default setting of $r = 8$, leading to an approximate token reduction ratio of 38%, we configure all compared methods to use the same token reduction ratio, so that their accuracy and computational overhead can be directly compared under aligned compression constraints. As shown in Table 3, SRE-Merge achieves the best retrieval accuracy on all three ReID benchmarks while requiring the lowest GFLOPs. These results suggest that generic token reduction methods designed for preserving global semantics in VLMs/MLLMs may overlook subtle identity-related cues in fine-grained retrieval, whereas SRE-Merge better preserves discriminative local details through task-aware spatial-entropic token merging.

To more comprehensively validate the advantages of SRE-Merge, we perform comparative experiments between SRE-Merge and state-of-the-art ReID methods on Market-1501 and MSMT17. SRE-Merge uses the default setting ($r$=8) and reduces GFLOPs by 23.9% compared to TransReID, while other methods do not perform compression. As shown in Table 4, SRE-Merge achieves competitive performance on both datasets. This is attributed to the fact that SES-Assess fully considers the semantic importance of tokens, thus retaining the crucial retrieval cues for ReID. We also conduct experiments on Occluded-Duke. As shown in Table 5, SRE-Merge achieves the best mAP and competitive Rank-1 among the compared methods, demonstrating its effectiveness in learning discriminative identity-related regions under occlusion.

*Table 7.* Ablation studies on Market-1501 to evaluate the effectiveness of proposed components.

| Index | Components | | | | Market-1501 | |
|---|---|---|---|---|---|---|
| | Baseline | SES-Assess | HCA-Match | EPW-Fuse | Rank-1 ↑ | mAP ↑ |
| 1 | ✓ | - | - | - | 94.9 | 88.6 |
| 2 | ✓ | ✓ | ✓ | - | 95.4 | 88.9 |
| 3 | ✓ | ✓ | - | ✓ | 95.5 | 89.1 |
| 4 | ✓ | ✓ | ✓ | ✓ | **95.8** | **89.2** |

*Table 8.* Sensitivity analysis of balancing coefficients $\delta_1$ and $\delta_2$ in HCA-Function on Market-1501.

*(a)* Balancing coefficient - $\delta_1$

| $\delta_1$ | 1 | 1/5 | 1/10 | 1/20 | 1/40 |
|---|---|---|---|---|---|
| Rank-1 | **95.8** | 95.5 | 95.2 | 95.2 | 94.8 |
| mAP | **89.2** | 88.7 | 88.5 | 88.1 | 87.7 |

*(b)* Balancing coefficient - $\delta_2$

| $\delta_2$ | 1 | 1/5 | 1/10 | 1/20 | 1/40 |
|---|---|---|---|---|---|
| Rank-1 | 95.2 | 95.5 | 95.5 | **95.8** | 95.6 |
| mAP | 88.1 | 88.8 | 89.0 | **89.2** | 89.0 |

To demonstrate the generalization ability of SRE-Merge, we also conduct experiments on the ImageNet-1k dataset. Table 6 compares the Rank-1 of various state-of-the-art token compression methods. When using the same backbone networks as other methods (i.e., DeiT-S, DeiT-T, and DeiT-B), SRE-Merge achieves the highest accuracy in all cases. This indicates that our method is not only applicable to the ReID task but also able to achieve favorable performance in the classification task.

### 4.4. Ablation Study

To evaluate the effectiveness of key components in our SRE-Merge, we conduct a series of ablation studies on the Market-1501 dataset. ToMe is adopted as the baseline method. The evaluated components include the SES-Assess strategy, the HCA-Match scheme, and the EPW-Fuse paradigm, with results summarized in Table 7. Progressive component additions yield consistent metric gains, validating each module's contribution to feature representation and retrieval accuracy. The full model (Index 4) achieves optimal performance, demonstrating the cumulative effectiveness of integrating SES-Assess, HCA-Match, and EPW-Fuse.

The balancing coefficients are designed to regulate the relative importance of semantic similarity and contextual influence in the HCA-Function. To investigate their impact on model performance, we conduct comparative experiments on the Market-1501 dataset under two distinct settings. As shown in Table 8, decreasing $\delta_1$ consistently leads to performance degradation, suggesting that semantic similarity should maintain a sufficiently strong role in the HCA-Function. For $\delta_2$, the performance first improves as $\delta_2$ decreases from 1 to 1/20, and then slightly drops when it further decreases to 1/40. The best performance is achieved with $\delta_1 = 1$ and $\delta_2 = 1/20$.

## 5. Conclusion

This paper presents **SRE-Merge**, a novel training-free framework that introduces spatially-regularized entropy into token merging for fine-grained ReID. By leveraging **SES-Assess**, we quantify token importance as **SE-Mass** to precisely identify discriminative regions. Furthermore, the **HCA-Match** and **EPW-Fuse** strategies utilize mass-derived context and variance-preserving constraints to prevent semantic aliasing and counteract feature variance reduction. Our method achieves a 24% reduction in GFLOPs while retaining competitive accuracy against state-of-the-art methods, establishing an effective accuracy-efficiency trade-off for lightweight ReID.

## Acknowledgements

This work was partially supported by the National Natural Science Foundation of China (No. 62472016) and the Fundamental Research Funds for the Central Universities.

## Impact Statement

This paper presents work whose goal is to advance the efficiency of Vision Transformers in fine-grained retrieval tasks. The proposed method, SRE-Merge, is evaluated on public academic datasets (e.g., Market-1501, MSMT17) that are widely used in the research community. While our work focuses on algorithmic acceleration and does not directly involve the collection of new data, we acknowledge that person re-identification technologies have potential societal implications regarding privacy and surveillance. We advocate for the responsible deployment of such technologies in strict compliance with local laws and privacy regulations. There are no other specific ethical issues that must be highlighted here.

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

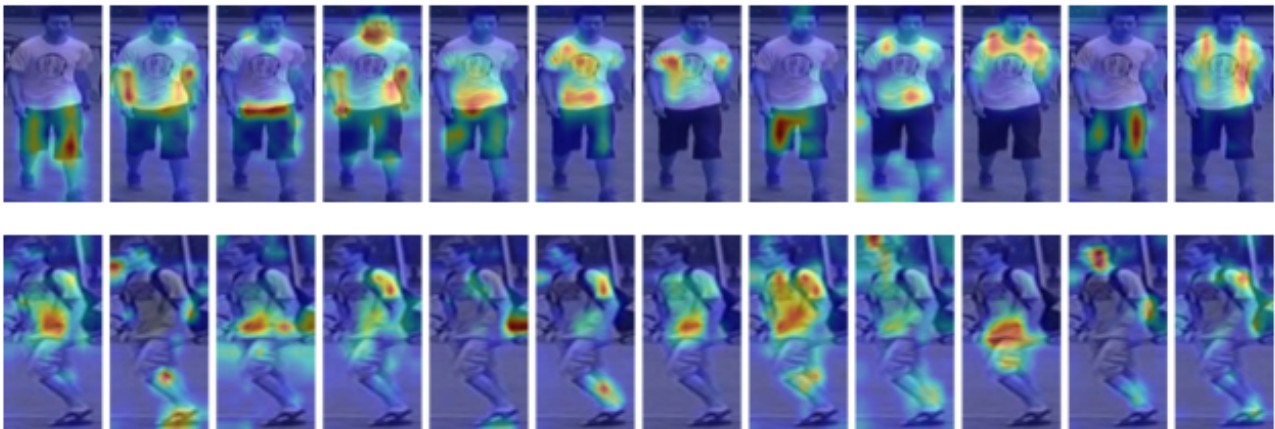

*Figure 3.* Spatial distribution of attentional foci across distinct attention heads. Each row corresponds to an individual image sample, with the left-to-right sequence demonstrating the 12 attention heads in the fourth transformer layer of TransReID.

## A. Analysis

### A.1. Rationale for differential weighting across attention heads

Existing token assessment methods typically compute either similarity scores with the class token or the sum of attention weights allocated by other query tokens when serving as a key. Then they average the outputs of these attention heads as the importance of the current token. However, individual heads usually exhibit differential responses towards different patterns. The simple averaging operation causes the loss of this diversity, leading to systemic underestimation of crucial tokens located in critical local regions.

To illustrate this issue, we visualize the spatial distributions of attentional foci across distinct attention heads. As can be seen in Figure 3, it demonstrates divergent spatial foci across heads: selective focus on foreground anatomical regions (e.g., head and limbs), appendages (e.g., backpack), and incidental background elements. This visualization conclusively validates the suboptimal nature of direct averaging across attention heads. Therefore, we propose the dynamic spatial-aware entropy weighted token assessment (SES-Assess) strategy to assign higher importance to tokens in critical local regions beneficial for identity discrimination.

### A.2. Rationale for utilizing SE-Mass during token merging.

Traditional token size-based methods adopt weighted averaging strategies to accomplish token merging. We point out that these strategies induce feature attenuation by neglecting contribution disparity among different tokens in identity matching. This problem becomes particularly pronounced when merging multiple tokens simultaneously.

To substantiate this claim, Figure 4 quantitatively illustrates the inverse correlation between average cosine similarity and token merging quantity. It demonstrates that the average cosine similarity progressively declines as the number of tokens to be merged increases. This downward trend indicates that merging an excessive number of tokens in a single operation may force the fusion of semantically divergent tokens, thereby obscuring the salient expression of critical information. Therefore, we incorporate SE-Mass into the weighting strategy to mitigate the shortcomings of token size-based methods and preserve feature discriminability.

### A.3. Rationality analysis of SES-Assess

As formalized in Equation (9), our token assessment strategy (SES-Assess) employs SE-Mass to quantify the importance of each token. This dual-gating mechanism requires both high class-token attention score and attention head weight to assign significance to a token. Even if some attention heads exhibit high activations on redundant regions in terms of class-token attention scores, such tokens are suppressed during SE-Mass computation when they are associated with high local neighborhood entropy or low spatial prior. This composite mechanism relies solely on the outputs of multi-head self-attention and requires no additional supervision, yet effectively identifies identity-discriminative tokens to enhance the retrieval ability of the ReID model.

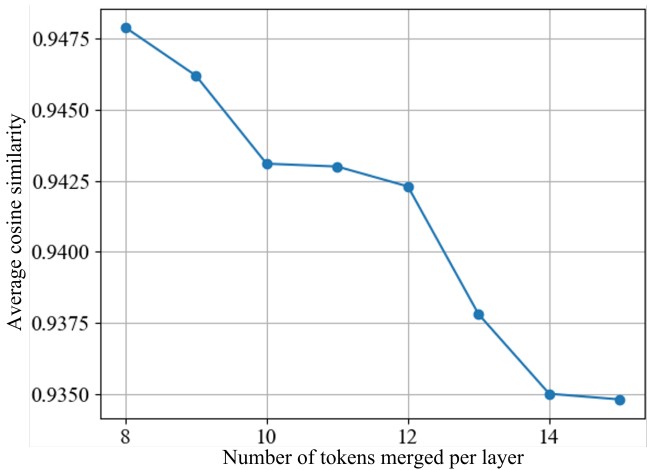

*Figure 4.* The relationship between average cosine similarity and token merging quantity.

---

**Algorithm 1** HCA-Match

---

**Input:** The Key $K$, SE-Mass $\mathcal{S}$, number of merged tokens $r$, balancing coefficients $\delta_1$ and $\delta_2$

**Output:** Indices of unmatched token pairs $T_{ui}$, reserved indices of matched token pairs $T_{si}$ and $T_{di}$, mass-ordered bipartite index sets $Index_A$ and $Index_B$

1: $sortScore \leftarrow \mathrm{argsort}(\mathcal{S}, \mathrm{order} = \mathrm{descending})$
2: $Index_A \leftarrow sortScore_{\mathrm{even}}$
3: $Index_B \leftarrow sortScore_{\mathrm{odd}}$
4: $\hat{\mathcal{S}} \leftarrow \frac{1/\mathcal{S}}{\max_k(1/\mathcal{S}_k)}$
5: $K_A \leftarrow K[Index_A], \quad K_B \leftarrow K[Index_B]$
6: $\mathcal{S}_A \leftarrow \hat{\mathcal{S}}[Index_A], \quad \mathcal{S}_B \leftarrow \hat{\mathcal{S}}[Index_B]$
7: $F_{sim} \leftarrow \frac{1}{2}\left(\frac{K_A \cdot K_B^T}{\|K_A\|_2 \cdot \|K_B\|_2} + 1\right)$
8: $F_{res} \leftarrow \mathcal{S}_A \times \mathcal{S}_B^T$
9: $Score \leftarrow F_{sim}^{\delta_1} \odot F_{res}^{\delta_2}$
10: $M_v \leftarrow \max(Score, \dim = 1)$
11: $M_i \leftarrow \mathrm{argmax}(Score, \dim = 1)$
12: $M_{vs} \leftarrow \mathrm{argsort}(M_v, \mathrm{order} = \mathrm{descending})$
13: $T_{si} \leftarrow M_{vs}[:r]$
14: $T_{di} \leftarrow M_i[T_{si}]$
15: $T_{ui} \leftarrow M_{vs}[r:]$

---

### A.4. Explanation of training-free property

In this paper, training-free means all mechanisms of SRE-Merge are implemented based on pre-trained models, without introducing new learnable parameters or requiring additional training, fine-tuning, or distillation of the models. Token merging via designed strategies is completed only during the inference phase. The implementation of SES-Assess, HCA-Match, and EPW-Fuse all rely on existing attention weights (e.g., outputs of multi-head self-attention) and feature vectors (e.g., K vectors) of the pre-trained models. All operations are parameter-free computations, involving no training updates. This design allows SRE-Merge to be directly deployed on pre-trained models, avoiding extra training costs, and is particularly suitable for resource-constrained real-world scenarios.

## B. Algorithms

The details of our proposed HCA-Match scheme and EPW-Fuse paradigm (with a regular merge operation) are described in Algorithms 1, 2, and 3.

---

**Algorithm 2** Merge

---

**Input:** Token sequence $T$, indices of unmatched token pairs $T_{ui}$, reserved indices of matched token pairs $T_{si}$ and $T_{di}$, aggregation operation $mode$, mass-ordered bipartite index sets $Index_A$ and $Index_B$
**Output:** Merged token sequence $T_m$
 1: $T_{\text{even}} \leftarrow T[Index_A], \quad T_{\text{odd}} \leftarrow T[Index_B]$
 2: $T_u \leftarrow T_{\text{even}}[T_{ui}]$
 3: $T_s \leftarrow T_{\text{even}}[T_{si}]$
 4: $T_d \leftarrow T_{\text{odd}}.\,\text{scatter\_reduce}(T_{di}, T_s, mode)$
 5: $T_m \leftarrow \text{concatenate}(T_u, T_d, \dim = 0)$

---

**Algorithm 3** EPW-Fuse

---

**Input:** Token sequence $Z$, SE-Mass $\mathcal{S}$, size of token sequence $Size$
**Output:** Merged token sequence $Z_m$
 1: $Z_{\text{score}} \leftarrow \text{Merge}(Z \times \mathcal{S}, mode = \text{sum})$
 2: $\mathcal{S} \leftarrow \text{Merge}(\mathcal{S}, mode = \text{sum})$
 3: $Size \leftarrow \text{Merge}(Size, mode = \text{sum})$
 4: $Z_{\text{score}} \leftarrow Z_{\text{score}}/\mathcal{S}$
 5: $Z_{\text{norm}} \leftarrow \|Z\|_2$
 6: $Z_{\text{score\_norm}} \leftarrow \|Z_{\text{score}}\|_2$
 7: $Z_{\text{norm\_max}} \leftarrow \text{Merge}(Z_{\text{norm}}, mode = \text{amax})$
 8: $Z_m \leftarrow \frac{Z_{\text{score}}}{Z_{\text{score\_norm}}} \times Z_{\text{norm\_max}}$

---

*Table 9.* Rank-1 (%) and mAP (%) under varying $k$ values on Market-1501.

| $k$ | 1 | 4 | 9 | 16 | 25 |
|---|---|---|---|---|---|
| Rank-1 | 95.3 | 95.4 | **95.8** | 95.4 | 95.2 |
| mAP | 88.8 | 89.0 | **89.2** | 89.1 | 89.0 |

## C. Experiments

### C.1. The relationship between GFLOPs and token merging quantity

Our experiments demonstrate the quantitative relationship between the number of merged tokens per layer ($r$) and GFLOPs (with merging applied only to the first 10 layers to preserve performance), as shown in Figure 5. Increasing $r$ progressively reduces GFLOPs, exhibiting an approximately linear correlation. This confirms that the token merging quantity directly dictates computational complexity. Such parametric control enables effective trade-offs between model performance and computational costs, providing theoretical foundations for optimizing inference efficiency across deployment scenarios.

### C.2. Test of the number of nearest tokens within a neighborhood

We further investigate the effect of the number of nearest neighbor tokens $k$ within each local neighborhood, as reported in Table 9. When $k$ is relatively small, the model may only capture limited local contextual information, which restricts the effectiveness of neighborhood-aware token matching. As $k$ increases, both Rank-1 and mAP show consistent improvements, indicating that incorporating more neighboring tokens helps the model better estimate contextual relevance. However, when $k$ becomes too large, the performance starts to decline, probably because excessive neighbors may introduce redundant or less relevant tokens and weaken the locality of the matching process. The best performance is achieved when $k$=9. Therefore, we set $k$=9 across all experimental configurations.

### C.3. Comparison with ToMe under varying GFLOPs on Market-1501

We conduct comparative experiments with ToMe on the Market-1501 dataset to analyze performance disparities under varying GFLOPs. As illustrated in Figure 6, SRE-Merge demonstrates consistent performance superiority across different computational budgets, attributable to our method's refined perception of critical local regions.

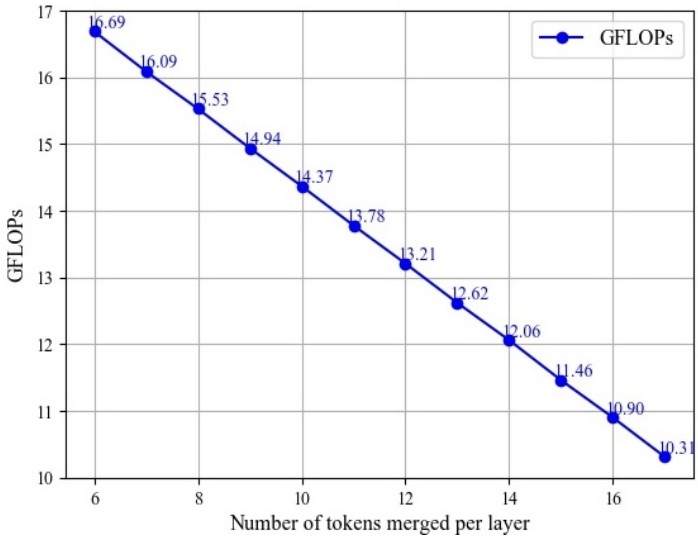

*Figure 5.* The relationship between GFLOPs and number of merged tokens $r$.

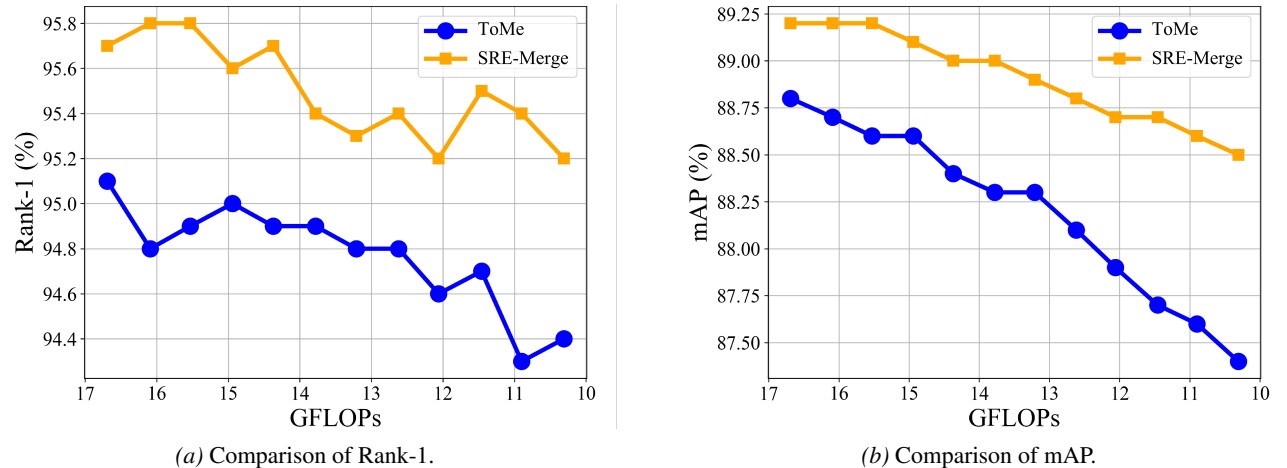

*(a)* Comparison of Rank-1.

*(b)* Comparison of mAP.

*Figure 6.* Comparison with ToMe under varying GFLOPs on Market-1501.

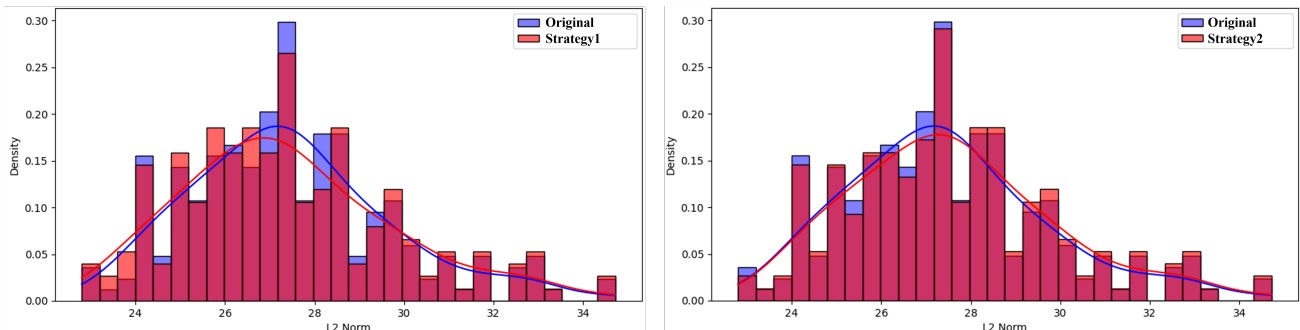

*Figure 7.* The $L2$ norm distributions of features before and after fusion using different strategies on Market-1501. Strategy 1 denotes token fusion through token size-based weighted averaging, while strategy 2 denotes our SE-Mass weighted token fusion with norm interpolation.

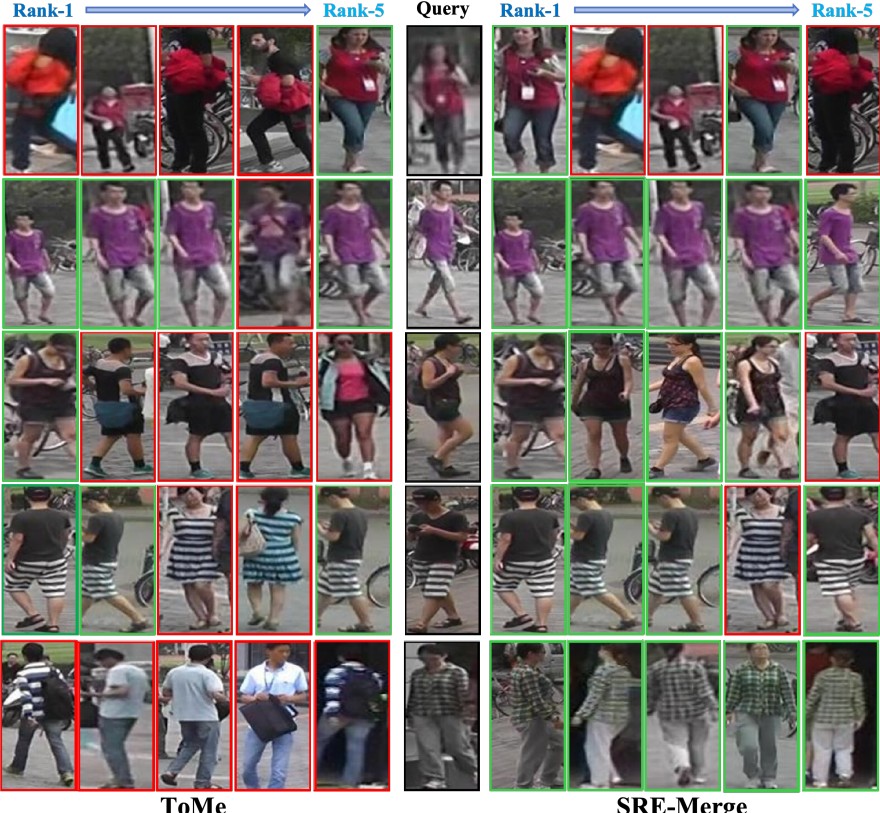

*Figure 8.* Visualization of retrieval results. Red boxes indicate incorrect retrieval results, while green boxes highlight correct matches.

## D. Visualization

### D.1. Visualization of L2 Norm Distributions Under Different Strategies

Taking the fourth layer of the model as an example, Figure 7 illustrates the $L2$ norm distributions of features using different strategies on Market-1501. "Original" refers to the norm distribution of raw features before fusion, shown in blue. "Strategy1" corresponds to the result of traditional token size-based weighted averaging, and "Strategy2" corresponds to our EPW-Fuse. Both Strategy1 and Strategy2 are shown in light red in both subfigures (areas overlaid on the blue background of the original distribution appear dark red). "Density" here refers to the probability density of feature norms, quantifying the proportion of samples with different norm values. High-norm regions (right side of each subfigure) usually correspond to highly discriminative key features (e.g., clothing patterns, limb details), while low-norm regions (left side) mostly represent background or redundant information. Notably, compared with the original norm distribution, Strategy1 exhibits marked contraction in high-norm regions post-fusion. The global downward shift of feature norms indicates energy dissipation in original features, potentially compromising identity discriminability. The distribution shows that the density of high-norm regions in Strategy1 decreases significantly, indicating that traditional fusion causes norm diminishment of key features; in contrast, the distribution of Strategy2 is closer to Original, especially with high-norm regions preserved, confirming the effectiveness of EPW-Fuse in alleviating norm diminishment.

### D.2. Visualization of retrieval results on Market-1501

To provide an intuitive evaluation of our method, we conduct visualization experiments comparing ToMe and SRE-Merge on the Market-1501 dataset, as illustrated in Figure 8. Specifically, given a query image, we retrieve the top-5 gallery images ranked by similarity scores. The visual results demonstrate that our SRE-Merge exhibits superior robustness compared to ToMe across diverse scenarios. With the proposed token merging mechanism, SRE-Merge effectively focuses on critical local regions and achieves good retrieval results.

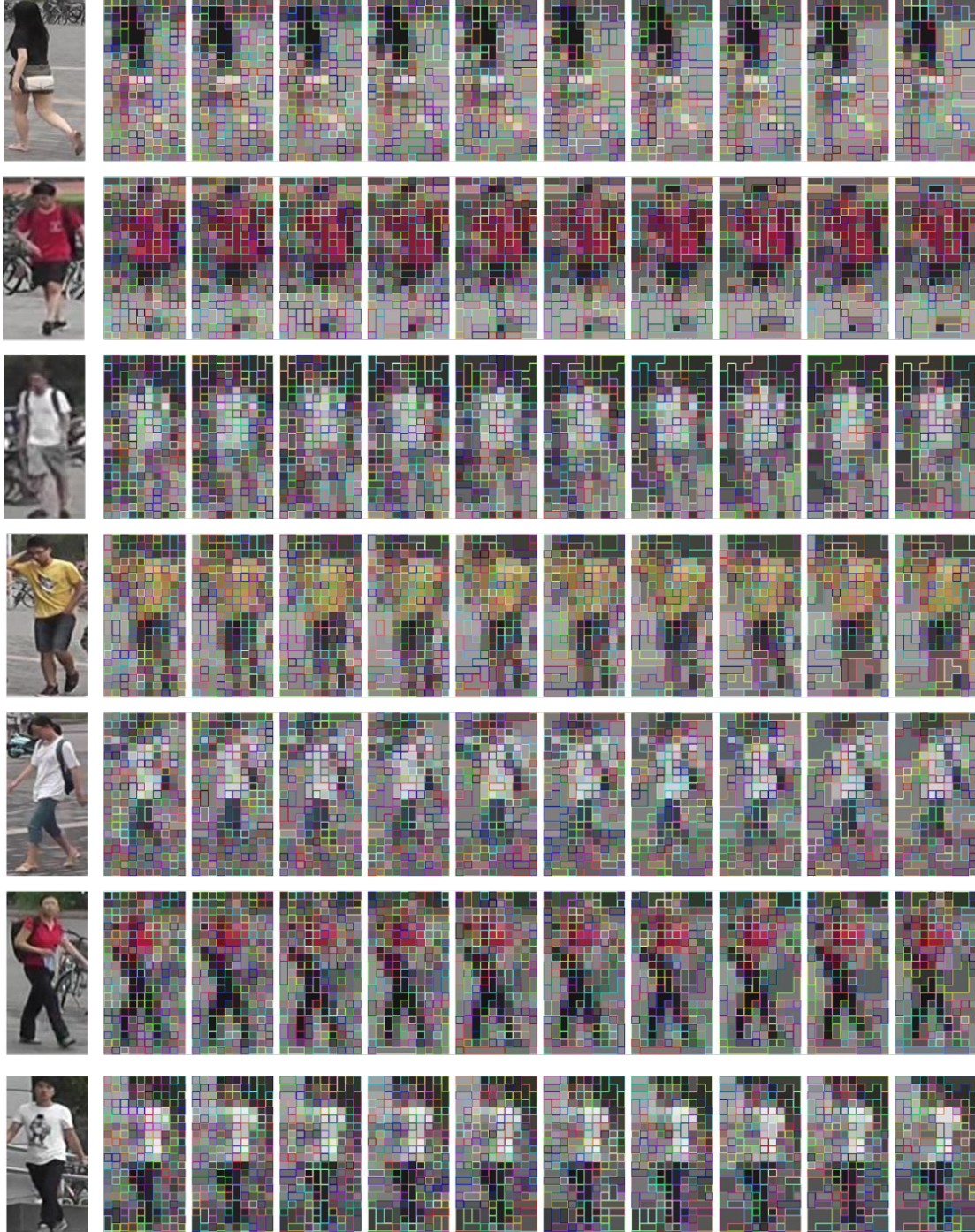

*Figure 9.* Visualization of token merging results. SRE-Merge progressively merges background or semantically homogeneous regions across layers while preserving identity-sensitive foreground details.

## D.3. Visualization of token merging results on Market-1501

We conduct visualization experiments demonstrating the compression effects across the first 10 layers, as shown in Figure 9. The number of merged tokens progressively increases as the layers deepen. Notably, our strategy predominantly merges tokens in background regions while effectively preserving foreground critical information.

