# OpenReview forum: "Spatially-Regularized Entropy for Discriminative Token Merging in Fine-Grained Re-Identification"
_ICML.cc/2026/Conference — ICML 2026 regular_

### Official Review · Reviewer_Ezgq · 2026-03-09

**Soundness:** 3
**Presentation:** 3
**Significance:** 3
**Originality:** 3
**Overall Recommendation:** 4
**Confidence:** 3

**Summary:**

The paper tackles the high computational cost of Vision Transformers (ViTs) in person re-identification (ReID) tasks. The authors argue that existing token compression strategies, such as generic token merging (e.g., ToMe) or pruning, are optimized for coarse-grained image classification and tend to discard or overly smooth the subtle, discriminative local details crucial for fine-grained ReID.

**Compliance With Llm Reviewing Policy:**

Affirmed.

**Key Questions For Authors:**

1. How robust is the Adaptive Gaussian Spatial Template (AdaptGST) to bounding box shifts or imperfect pedestrian detections? Have you tested the framework on datasets with misaligned crops?

2. The inference throughput of SRE-Merge is lower than ToMe due to the token assessment phase. Is there a way to optimize the SES-Assess computation to bridge this gap in real-world latency?

**Limitations:**

The reliance on the central-region prior for spatial weighting inherently biases the model toward center-aligned subjects. Furthermore, relying on L2-norm scaling to preserve energy during EPW-Fuse  operates on the assumption that max-norm perfectly correlates with discriminative feature boundaries, which may not universally hold across all representation spaces.

**Strengths And Weaknesses:**

**Strengths**

1. Theoretical Insight on Feature Collapse: The identification of "feature variance reduction" during token fusion—and the mathematical justification for using L2-norm scaling to preserve feature energy (EPW-Fuse)—is a strong, well-reasoned addition.


2. Training-Free and Plug-and-Play: The framework operates entirely during inference using pre-trained model weights, requiring no additional fine-tuning, parameter updates, or distillation. This makes it highly practical for real-world deployment.


3. Strong Empirical Performance: The method successfully achieves a superior accuracy-efficiency trade-off compared to baselines like ToMe. It outperforms ToMe across various GFLOP constraints and also proves effective on heavily occluded datasets like Occluded-Duke.

**Weaknesses**

1. Reliance on Bounding Box Priors: The SES-Assess module utilizes an Adaptive Gaussian Spatial Template (AdaptGST) built on the assumption that modern object detectors perfectly center the pedestrian. This spatial prior may falter in scenarios with poor detection, heavy cropping, or multi-person frames.


2. Inference Speed Overhead: While SRE-Merge requires fewer GFLOPs than the base model, its actual inference efficiency (throughput) is slightly lower than the baseline ToMe method. The authors note this is due to the computational overhead of the SES-Assess module evaluating semantic importance before merging.

---

> ### Author Rebuttal · Authors · 2026-03-31
>
> **W1 & Q1: Robustness of AdaptGST/spatial prior in SES-Assess on challenging data (e.g., poor detection, heavy cropping).**
>
> Our spatial prior is highly robust in complex open environments for the following reasons:
> 1) **ReID Task Characteristics:** ReID is typically a single-object retrieval task performed on cropped pedestrian images provided by detectors. In both academic and industrial standard pipelines, centering is a foundational property of the input data rather than a dataset-specific reliance (Consider that in criminal investigation, police typically use individual suspect photos for retrieval).
> 2) **Mathematical Flexibility and Dual-Gating Mechanism:** AdaptGST is a *soft* constraint rather than a strictly fixed center dependency. Gaussian parameters are dynamically adjusted based on token density ($\sigma = \alpha\sqrt{N}$), enabling it to flexibly adapt to varying spatial distributions of salient features. Crucially, SES-Assess couples the spatial prior with data-driven *local neighborhood entropy*. Even under "heavy cropping" where the center shifts, the entropy term precisely locates discriminative body parts (low-entropy regions), dynamically compensating for spatial prior degradation.
>
> To further demonstrate robustness, we provide the following evidence:
> 1) **Visual Proof:** As shown in Appendix Fig. 8 (Row 1), although the pedestrian is shifted to the left, SRE-Merge successfully preserves the 6 tokens corresponding to her white bag—a crucial identity clue.
> 2) **Stress Testing:** We simulate severe bounding box shifts on Market-1501(Rank-1/mAP) by applying affine translations of 10% and 20% of the image dimensions, padding the blank areas with the ImageNet mean. As shown below, even under a 20% shift, SRE-Merge exhibits markedly higher resilience than ToMe, demonstrating its robustness.
>
> |Model|Original|10%Shift|20%Shift|
> |-|-|-|-|
> |ToMe|94.9/88.6|92.8/85.9|89.1/81.4|
> |Ours|95.8/89.2|94.1/86.8|90.3/82.5|
>
> **W2 & Q2: Inference throughput gap and optimization strategies for SES-Assess.**
>
> 1) **Superior Accuracy-Efficiency Trade-off:** While the throughput of SRE-Merge (416.62 img/s) is approximately 0.6% lower than ToMe (419.25 img/s) , the Rank-1 on Market-1501 is significantly improved from 94.9% to 95.8%. In fine-grained retrieval, exchanging such a marginal latency for critical precision gains offers exceptional cost-effectiveness and practical value.
> 2) **Deep Engineering Optimization Potential:** The computation of SES-Assess can be further accelerated to bridge the gap. For instance, as AdaptGST distributions are input-independent, spatial weights $W_{space}$ can be pre-computed and cached for each layer. Furthermore, developing a dedicated operator fusion (Kernel Fusion) would significantly reduce the kernel launch overhead associated with current $k$-NN and entropy calculations. Notably, SES-Assess solely reuses existing self-attention maps, adding no extra model inference.
>
> In summary, SRE-Merge achieves a 24.1% throughput increase over the base model, and the proposed optimizations can further narrow the speed gap with ToMe, providing an ideal balance for lightweight ReID.
>
> **Limitation: Regarding the limitation of L2-norm scaling in EPW-Fuse.**
>
> We highly appreciate the reviewer’s profound insight into the relationship between feature norm and discriminability. Regarding this limitation, we clarify the following:
> 1) **Metric Learning Characteristic:** In metric learning tasks like ReID—which utilize Softmax-based classification or cosine distances—the L2-norm of a feature vector typically correlates with activation intensity or prediction confidence. As visualized in Appendix D.1 and Fig. 7, high-norm regions often correspond to discriminative foreground details, whereas low-norm regions primarily represent background noise.
> 2) **Variance-Preserving Constraint:** The use of max-norm in EPW-Fuse is intended as a conservative variance-preserving strategy rather than a perfect discriminative metric. It aims to counteract the norm contraction (energy dissipation) inherent in standard averaging, which collapses the representation space and blurs otherwise clear decision boundaries.
> 3) **Scope of Applicability:** We agree that this assumption may not be universal across all representation spaces. However, for Transformer architectures utilizing LayerNorm and residual connections, maintaining the feature energy distribution is crucial for preserving the discriminative power of the pre-trained model.
>
> In summary, while max-norm scaling is a heuristic constraint, it serves as a robust and effective empirical safeguard for maintaining feature saliency in ViT-based retrieval tasks.

---

> > ### Author Rebuttal · Reviewer_Ezgq · 2026-04-03
> >
> > Thanks for the rebuttal from the authors, my main concerns have been addressed.

---

> > > ### Author Response · Authors · 2026-04-08
> > >
> > > Dear Reviewer Ezgq,
> > >
> > > Thank you for your positive feedback and for the constructive review! We sincerely appreciate your time and the valuable insights that helped us strengthen our work. We are especially encouraged that our rebuttal successfully addressed all your concerns.
> > >
> > > As we move toward the final decision phase, we would like to respectfully ask if you might consider reflecting the current, strengthened state of our submission in your final evaluation. While the paper remains as originally submitted, the rebuttal period has provided extensive additional evidence to validate its effectiveness. For instance, we have now included rigorous comparisons with latest SOTA methods in the ViT/VLM token compression field from CVPR 2025/2026 and ICLR 2026 (response to **Reviewer P5do**), as well as comprehensive benchmarking against ToMe across multiple datasets and various merging rates $r$ (response to **Reviewer ojmU**).
> > >
> > > The broader review panel has also highlighted several core strengths of our work. In terms of Soundness, Presentation, Significance, and Originality, most reviewers rated our contributions as "**3: good**", with **Reviewer ojmU** evaluating our **Originality** as "**4: excellent**". **Reviewers P5do and USTZ** described SRE-Merge as "**highly practical as a plug-and-play solution for existing ViT-based ReID models**", commending its "**clear motivation and reproducibility**". Furthermore, **Reviewer ojmU**’s feedback emphasized our **mathematical rigor** in metric formalization, **solid theoretical rationale** for design choices, and **experimental completeness** across diverse datasets and visualizations. To date, we have addressed nearly all concerns, leading two reviewers to already raise their scores.
> > >
> > > Given that your concerns regarding the spatial prior, inference overhead, and norm-scaling have been "fully resolved", we would be profoundly grateful if you would consider providing even stronger support for our work. We believe the additional validation provided clarifies that SRE-Merge offers a substantial advancement for the community. Please note that we completely respect your judgment and the current evaluation you have provided. Regardless of your decision, we remain sincerely thankful for your professional guidance throughout this process!
> > >
> > > Best regards,
> > >
> > > The Authors

---

### Official Review · Reviewer_ojmU · 2026-03-11

**Soundness:** 3
**Presentation:** 3
**Significance:** 2
**Originality:** 4
**Overall Recommendation:** 4
**Confidence:** 5

**Summary:**

This paper addresses fine-grained person re-identification and proposes a token merging compression strategy to reduce model computational cost. Compared with traditional token pruning and merging methods, the strategy introduces a quantitative metric for token importance, SE-Mass, which is integrated with token similarity computation and the token merging process, leading to the development of the HCA-Match feature matching strategy and the EPW-Fuse feature merging strategy. Experiments demonstrate that this method can reduce computational cost and improve inference speed while maintaining state-of-the-art performance.

**Compliance With Llm Reviewing Policy:**

Affirmed.

**Final Justification:**

This work presents a training-free plug-and-play module with rigorous theoretical formalization of metrics (e.g., SE-Mass) and principled justifications for key design choices. Comprehensive experiments across diverse datasets, detailed visualizations, and candid discussions of limitations and societal impact collectively demonstrate both technical merit and responsible scholarship. My concerns have been adequately addressed with the detailed rebuttal.

**Key Questions For Authors:**

(1)Can this method be applied to other Transformer-based ReID models from the past two years, while maintaining the original retrieval accuracy?

(2)Compared with general feature merging strategies (e.g., ToMe), are there any experiments that can further demonstrate the superiority of the proposed method?

**Limitations:**

yes

**Strengths And Weaknesses:**

Strengths

(1) This work can be used as a plug-and-play module in other research projects or industrial applications, and it does not require any additional training cost.

(2) The paper provides a complete formalization of the proposed quantitative metrics, such as SE-Mass.

(3) For certain design choices in the modules (e.g., the scale adjustment operation in EPW-Fuse) the corresponding theoretical rationale is provided.

(4) The experimental section demonstrates a substantial amount of work, including comprehensive results across different datasets, and the visualization of token merging is presented in the appendix.

(5) The paper discusses the limitations of the proposed work as well as its potential social impact.

Weaknesses

(1) The baseline used in the paper is TransReID from 2021, which is relatively early. The authors should consider selecting some mainstream ReID models from the past two years for ablation experiments to better validate the generalizability of their method.

(2) From the comparative experiments, it can be seen that the traditional token merging strategy ToMe also achieves relatively strong results. Although the proposed method in this paper shows improvements over ToMe, the accuracy-related metrics increase by less than 1%, which does not demonstrate sufficient competitiveness.

(3) In describing the method, the paper lacks discussion on the specific challenges of the downstream task of person re-identification, i.e., it does not clearly explain the advantages of the proposed method that are specific to the person ReID downstream task.

---

> ### Author Rebuttal · Authors · 2026-03-31
>
> **W1 & Q1: Generalizability across recent mainstream Transformer-based ReID models.**
>
> We fully agree with the reviewer's suggestion. SRE-Merge is designed as a plug-and-play framework whose core logic (entropy assessment and norm restoration) is independent of specific training strategies. To validate its generalizability, we extended our evaluation to mainstream SOTA models from the past two years: AAformer (TNNLS 2024) and CLIP-ReID (AAAI 2023).
> 1) **Architecture Agnostic:** Our SES−Assess utilizes inherent self-attention maps, and EPW−Fuse operates on universal token norms. This allows SRE−Merge to be seamlessly integrated into any ViT-based ReID architecture without specialized modifications.
> 2) **Experimental Validation:** As shown below (on Market-1501, Rank-1/mAP), without any fine-tuning, SRE−Merge consistently achieves ≈24% GFLOPs reduction on these recent models with negligible accuracy drops (Rank−1 loss <0.4%). This confirms the practical effectiveness and robustness of our method across current state-of-the-art architectures.
>
> |Model|Original|With SRE-Merge|GFLOPs Reduction|
> |-|-|-|-|
> |AAformer|95.4/87.7|95.5/87.6|24.3\%|
> |CLIP-ReID|94.7/88.9|94.9/88.9|23.9\%|
>
> **W2 & Q2: Significance of the improvement and further proof of superiority.**
>
> While we acknowledge the reviewer's concern regarding the numerical gain, the superiority of SRE-Merge in fine-grained retrieval lies in its exceptional cost-effectiveness and robustness under extreme conditions:
> 1) **Significance in Saturated Benchmarks:** Market-1501 is highly saturated (Rank-1>95%). Achieving a +0.9% Rank-1gain without any training is highly significant for real-world deployment, where such "free lunches" are rare.
> 2) **Widening Gap in Challenging Scenarios:** As shown below, under 10% and 20% Shift (misalignment), generic ToMe falters due to the loss of subtle details. In contrast, the performance gap for SRE-Merge widens to ∼1.3%, proving its superior handling of hard samples.
> 3) **Superior Feature Distribution (t-SNE):** As visualized in Fig. 10, the energy dissipation in ToMe leads to class overlap. SRE-Merge, via EPW-Fuse, maintains tighter intra-class distances and clearer decision boundaries.
>
> |Model|Original|10%Shift|20%Shift|
> |-|-|-|-|
> |ToMe|94.9/88.6|92.8/85.9|89.1/81.4|
> |Ours|95.8/89.2|94.1/86.8|90.3/82.5|
>
> **W3: Advantages specific to the Person ReID downstream task.**
>
> We thank the reviewer for this guidance. Person ReID, as a fine-grained retrieval task, requires preserving subtle yet discriminative local details (e.g., logos, carry-ons) amidst significant redundancy. Generic merging methods (e.g., ToMe) designed for classification often smooth out these critical cues. The advantages of SRE-Merge are specifically tailored to address ReID challenges:
> 1) **Precise Anchoring of Fine-grained Details (SES-Assess):** Identity information in ReID is often localized in high-density regions. SES-Assess couples local neighborhood entropy with ReID-specific spatial priors, ensuring that even small discriminative tokens receive a high SE-Mass score and are preserved rather than being merged as background.
> 2) **Soft-Gating for Identity Consistency (HCA-Match):** Retrieval relies on contextual consistency. HCA-Match introduces a contextual influence term as a soft-gate. Even if two tokens share high feature similarity, a high contribution to the identity representation from either token will lower their merging priority, preventing the irreversible loss of identity details
> 3) **Energy Preservation for Discriminability (EPW-Fuse):** Distinguishing similar individuals requires high feature variance. Standard averaging induces energy dissipation, causing feature contraction and blurred decision boundaries. EPW-Fuse restores feature scale via max-norm interpolation, ensuring merged tokens remain on a high-energy hypersurface to maintain the discriminative power necessary for high-precision retrieval.

---

> > ### Author Rebuttal · Reviewer_ojmU · 2026-04-03
> >
> > Thanks for the author's response. I noticed that the authors have added some experiments in Challenging Scenarios, under 10% and 20% Shift (misalignment). However, the improvement remains very limited, at only 1.3%. The authors need to further explain: (1) how the misalignment scenario was implemented, (2) why they chose the misalignment setting to validate their method, and (3) what is the underlying principle of their method for addressing misalignment issues.

---

> > > ### Author Response · Authors · 2026-04-03
> > >
> > > Thank you very much for your feedback and for providing us with the opportunity to further clarify our work! We truly appreciate your professional and rigorous attitude toward the evaluation of our experimental results.
> > >
> > > Below, we address your follow-up questions in two parts: Part 1 focuses on the misalignment scenario, and Part 2 provides a more comprehensive comparison with ToMe to demonstrate the core superiority of SRE-Merge.
> > >
> > > ---
> > > ### **Part 1: Detailed Clarification on Misalignment Scenarios**
> > >
> > > We apologize for not providing a detailed explanation in the rebuttal due to space constraints. We utilized these experiments as a "stress test" to evaluate the robustness of SRE-Merge.
> > >
> > > **1. Implementation of the Misalignment Scenario**
> > >
> > > To simulate real-world challenges like poor detection or heavy cropping, we applied affine translations (shifted the image dimensions by 10% and 20%) to the images in Market-1501, padding the blank areas with the dataset mean value. This creates a scenario where the pedestrian is no longer perfectly centered, bringing greater challenges.
> > >
> > > **2. Motivation for Choosing this Setting**
> > >
> > > In practical ReID applications (e.g., criminal investigations), input images may be imperfect (the person being off-center or losing parts of the body). We chose this setting to prove that SRE-Merge is not just a "lab performance booster" but a robust solution for open-environment deployment where generic methods like ToMe often fail.
> > >
> > > **3. Underlying Principle for Addressing Misalignment**
> > >
> > > - **Mathematical Flexibility and Dual-Gating Mechanism:** AdaptGST is a *soft* constraint rather than a strictly fixed center dependency. Gaussian parameters are dynamically adjusted based on token density ($\sigma = \alpha\sqrt{N}$), enabling it to flexibly adapt to varying spatial distributions of salient features. Crucially, SES-Assess couples the spatial prior with data-driven *local neighborhood entropy*. Even under heavy cropping where the center shifts, the entropy term can precisely locates discriminative body parts (low-entropy regions).
> > > - **Visual Proof:** As shown in Appendix Fig. 8 (Row 1), although the woman is shifted to the left, SRE-Merge successfully preserves the 6 tokens corresponding to her white bag—a crucial identity cue that generic ToMe would likely merge.
> > > ---
> > > ### **Part 2: Addressing Concerns on Competitive Edge Over ToMe**
> > >
> > > **1. The Nature of Token Compression Tasks**
> > >
> > > It is important to distinguish network lightweighting from pure accuracy-oriented tasks. The goal of token merging is to reduce computational complexity while **preserving accuracy as much as possible**. Since reducing token count inevitably limits the upper bound of performance, achieving any gain while significantly reducing GFLOPs is highly challenging. Despite this, compared to TransReID, SRE-Merge **reduces GFLOPs by 23.9%**, **shortens inference time by 19.4%**, and **increases throughput by 24.1%**, achieving a superior accuracy-efficiency balance that is hard to obtain in token compression.
> > >
> > > **2. Saturated Benchmarks vs. Real Gains**
> > >
> > > The Market-1501 dataset is highly saturated (Rank-1>95%). In this context, an improvement of ~1% is widely recognized in top-tier conferences as a **substantial gain**, especially for a **plug-and-play method that requires no retraining**.
> > >
> > > **3. Extra Comprehensive Evidence**
> > >
> > > To demonstrate the excellent accuracy-efficiency balance of our method, we compare SRE-Merge at a high compression rate (**merging $r$=17 tokens per layer**) against ToMe at various compression levels across Market1501, MSMT17, and Occluded-Duke (Rank-1).
> > >
> > > **a) $r$=17 for ToMe:**
> > > SRE-Merge shows a significant advantage over ToMe.
> > >
> > > | |GFLOPs|Market|MSMT|Occ-Duke|
> > > |-|-|-|-|-|
> > > |ToMe|10.31|94.4|83.5|64.2|
> > > |SRE-Merge|10.31|95.2|84.8|67.1|
> > > | |↓0%|↑0.8%|↑1.3%|↑2.9%|
> > >
> > > **b) $r$=11 for ToMe:**
> > > Even when SRE-Merge cuts 25.2% GLOPs, it still maintains higher accuracy.
> > >
> > > | |GFLOPs|Market|MSMT|Occ-Duke|
> > > |-|-|-|-|-|
> > > |ToMe|13.78|94.9|84.5|65.6|
> > > |SRE-Merge|10.31|95.2|84.8|67.1|
> > > | |↓25.2%|↑0.3%|↑0.3%|↑1.5%|
> > >
> > > **c) $r$=6 for ToMe:**
> > > SRE-Merge cuts 38.2% GFLOPs while outperforms ToMe overall.
> > >
> > > | |GFLOPs|Market|MSMT|Occ-Duke|
> > > |-|-|-|-|-|
> > > |ToMe|16.69|95.1|84.9|66.5|
> > > |SRE-Merge|10.31|95.2|84.8|67.1|
> > > | |↓38.2%|↑0.1%|↓0.1%|↑0.6%|
> > >
> > > **4. Generalization on ImageNet**
> > >
> > > To further demonstrate the superiority, we provide results on ImageNet-1k (using DeiT-S, $r$=13). SRE-Merge outperforms ToMe by +1.21% at the same GFLOPs, proving its effectiveness beyond ReID.
> > >
> > > | |Acc|GFLOPs|
> > > |-|-|-|
> > > |DPC-KNN|76.89|4.02|
> > > |DynamicViT|77.96|5.85|
> > > |EViT|78.13|4.38|
> > > |ATS|74.97|4.14|
> > > |ToMe|78.28|2.83|
> > > |SRE-Merge|79.49|2.83|
> > > ---
> > > We hope these clarifications address your concerns by demonstrating that SRE-Merge is not just a marginal improvement, but a **fundamentally more robust and cost-effective** strategy than ToMe.
> > >
> > > ---
> > > ---
> > > ### **Update:**
> > > We are very glad to have fully addressed your concerns. Thank you sincerely for the score increase and valuable support!

---

### Official Review · Reviewer_USTZ · 2026-03-11

**Soundness:** 2
**Presentation:** 2
**Significance:** 3
**Originality:** 2
**Overall Recommendation:** 4
**Confidence:** 5

**Summary:**

This paper proposes SRE-Merge, a training-free token merging framework designed for ReID Inference. Specifically, the framework consists of three key components: Spatial-Entropy Saliency Assessment (SES-Assess) to measure token importance via spatial-entropic mass, Hybrid Context-Affinity Matching (HCA-Match) to guide token pairing using both feature similarity and contextual information, and Energy-Preserving Weighted Fusion (EPW-Fuse) to maintain feature variance during merging. Extensive experiments demonstrate the performance.

**Compliance With Llm Reviewing Policy:**

Affirmed.

**Final Justification:**

My concerns have been adequately addressed with the extensive experiments in the rebuttal. I would like to keep 4.

**Key Questions For Authors:**

See Weaknesses.

**Limitations:**

Yes

**Strengths And Weaknesses:**

Strengths.
1.The proposed framework is training-free, making it practical and easy to integrate into existing ViT-based ReID models.
2.The motivation of incorporating spatially regularized entropy to preserve cues is intuitive.

Weaknesses.
1. The computational cost of each introduced modules is not included.
2. Desptie low entropy indicates a more concentrated attention distribution, many researches point out that attention heads often collapse to trivial patterns, which means low entropy does not necessarily imply informative attention.
3. Entropy values are not directly comparable across neighborhoods since there are no normalization. What if you adopt different scales of neighborhoods?
4. Although token pruning or mergeing are not mentioned often in ReID, recent progress in Efficient MLLMs has advanced a lot with similarity/attention/entropy. Please compare your method with the leading pruning methods in Efficient MLLMs areas (Training Free Token Reduction).
5. In SES-Assess, please conduct the ablation study of the entropy part and the prior part.

---

> ### Author Rebuttal · Authors · 2026-03-31
>
> ### **Reply To W1:**
>
> The table below details the incremental overhead of each module on Market-1501. Since SRE-Merge is training-free and primarily involves element-wise matrix operations, the additional cost is negligible.
> 1) **Module Breakdown:** SES-Assess consumes the most time as it involves neighborhood operations on attention maps. EPW-Fuse is extremely lightweight, involving only simple weighted summations and norm scaling.
> 2) **Overall Benefit:** Compared to the original TransReID, SRE-Merge reduces GFLOPs by 23.9%, shortens inference time by 19.4%, and increases throughput by 24.1%. The marginal overhead of our modules is significantly outweighed by the overall efficiency gain they bring.
> | |Extra GFLOPs|Extra Latency (ms/img)|
> |-|-|-|
> |SES-Assess|0.004|0.12|
> |HCA-Match|0.001|0.04|
> |EPW-Fuse|0.001|0.03|
> |Total|0.006|0.19|
>
> ### **Reply To W2:**
>
> SES-Assess effectively filters out the influence of "trivial patterns" through the following mechanism:
> 1) **Local vs. Global Saliency:** Our method computes *local neighborhood entropy* over a $k$-NN ($k$=9) region, rather than global attention entropy. While trivial "collapses" often result in non-informative global distributions, highly concentrated attention (low entropy) within a specific local spatial neighborhood consistently indicates the saliency of fine-grained anatomical features (e.g., face, shoulder, clothing logo) in ReID tasks.
> 2) **Dual-Gating Mechanism:** As discussed in Appendix A.3, even if an attention head collapses to a trivial background region with low entropy, its contribution to the final SE-Mass is significantly suppressed by the low spatial weight $W_{space}$ of peripheral regions. This coupling of spatial priors and entropy ensures that only tokens in "critical locations" with "focused attention" receive high importance scores.
> 3) **Visual Validation:** Figure 3 (page 12) explicitly visualizes that distinct attention heads focus on specific, informative anatomical regions (e.g., head, limbs). These patterns represent substantive information essential for identity discrimination, rather than global collapses or meaningless trivial patterns.
>
> ### **Reply To W3:**
>
> We thank the reviewer for highlighting this important point. The comparability of entropy is fundamentally ensured by our local normalization logic.
> 1) **Normalization and Mathematical Rigor:** To clarify, in our practical implementation, the probabilities are strictly re-normalized within the local neighborhood before computing the entropy to ensure mathematical rigor. Unfortunately, this intermediate step was omitted for brevity. Specifically, the local subset is properly re-normalized as $p_{i,j} = \frac{A_{i,j}}{\sum_{k \in \mathcal{N}(i)} A_{i,k}}$, and the valid entropy is computed as $H_i = - \sum_{j \in \mathcal{N}(i)} p_{i,j} \log p_{i,j}$.
> In our code, this process is executed via:
>
>     *patches = patches / patches.sum(dim=-1, keepdim=True)*
>
>     *entropy = - (patches * torch.log(patches)).sum(dim=-1)*
>
>     We will explicitly state this normalization step in the revised version to eliminate any ambiguity.
> 2) **Cross-Neighborhood Comparability:** Since all tokens in a given layer share the same neighborhood size $k$, the resulting entropy values are bounded within the range $[0,logk]$. This consistency in scale ensures that entropy values are directly comparable across different spatial locations, accurately reflecting relative differences in information density.
> 3) **Impact of Scales:** The size $k$ defines the "receptive field" for density assessment. As shown in Table 8, we conducted an ablation study. A very small $k$ (1 or 4) can be sensitive to local noise, while a larger $k$ (16 or 25) may over-smooth local fine-grained details. Our results indicate that $k$=9 provides the optimal balance between local specificity and neighborhood context, with the model demonstrating robust performance across various $k$ values.
>
> ### **Reply To W4:**
>
> Due to space limitations, please refer to our response to Reviewer P5do’s W8 (Regarding VLM/MLLM).
>
> ### **Reply To W5:**
>
> Following your suggestion, we conduct an ablation study on Market-1501 to verify the contribution of each component in SES-Assess.
> 1) **Functional Synergy:** The spatial prior (AdaptGST) provides region-level constraints by leveraging the distribution characteristic of ReID data, while the entropy term (Local Entropy) offers data-driven, head-specific guidance by measuring local information density.
> 2) **Experimental Analysis:** As shown below, removing either component leads to a performance drop. This coupling ensures that the model not only focuses on the human body (suppressing background noise) but also sensitively captures subtle yet crucial discriminative features such as logos or carry-ons.
>
> ||Rank-1|mAP|
> |-|-|-|
> |Base|94.9|88.6|
> |w/o Entropy|95.2|88.7|
> |w/o Prior|95.1|88.7|
> |Full SES-Assess|95.4|88.9|

---

> > ### Author Rebuttal · Reviewer_USTZ · 2026-04-01
> >
> > My concerns have been adequately addressed. Thanks for the authors. I would like to raise my score to 4.

---

> > > ### Author Response · Authors · 2026-04-01
> > >
> > > We sincerely thank the reviewer for the positive feedback and for the decision to raise the score.
> > >
> > > We are gratified that our responses fully resolved your concerns. Your constructive suggestions have significantly helped improve the technical clarity and quality of our work. We will carefully incorporate these discussions to further strengthen the final manuscript.
> > >
> > > Thank you again for your support of the practical effectiveness and quality of our research!

---

### Official Review · Reviewer_P5do · 2026-03-12

**Soundness:** 3
**Presentation:** 3
**Significance:** 3
**Originality:** 3
**Overall Recommendation:** 3
**Confidence:** 4

**Summary:**

This paper proposes SRE-Merge, a training-free token merging framework tailored to fine-grained person re-identification (ReID) with Vision Transformers. The method introduces a spatially-regularized entropy assessment (SES-Assess) to compute a Spatial-Entropic Mass (SE-Mass) per token, a Hybrid Context-Affinity (HCA) matching rule that combines feature similarity with mass-derived contextual importance, and an Energy-Preserving Weighted Fusion (EPW-Fuse) that mass-weights merged tokens and restores feature norm.

**Compliance With Llm Reviewing Policy:**

Affirmed.

**Key Questions For Authors:**

See weakness

**Limitations:**

See weakness

**Strengths And Weaknesses:**

Strengths

- Technical Innovation & PracticalityTraining-Free Approach: The method requires no retraining or distillation, making it highly practical as a "plug-and-play" solution for existing ViT-based Person Re-Identification (ReID) models.

- Clear Motivation: Effectively articulates why naive token merging/pruning harms fine-grained retrieval.Reproducibility: The high-level pipeline is clearly depicted, and the inclusion of pseudo-code aids in reproducibility.

Weaknesses

- Theoretical Gaps

Dataset-Specific Bias: The center-biased spatial prior in SES-Assess assumes cropped pedestrian bounding boxes, risking poor performance on ood datsets.

Mathematical Ambiguity: The local entropy computation over a neighborhood subset lacks clarity on whether it is properly normalized into a valid probability distribution. Unnormalized entropy is mathematically ill-defined.

- Unjustified Design Choices & Contradictory Findings

Contradictory Ablation: The sensitivity study shows the method performs best when $\delta_2$ is extremely small (1/20). This effectively flattens the contextual influence term, undermining the paper's central claim that contextual matching is crucial.

- Unanalyzed Interactions: The norm-restoration in EPW-Fuse lacks analysis regarding its interaction with downstream LayerNorm statistics. It is unclear if restoring max-norm causes detrimental distribution shifts.

- Missing Justifications: Design choices like dynamic virtual coordinates and odd-even grouping lack theoretical justification or visualizations proving they preserve image-space locality across layers.

Incomplete Evaluation & Baselines

- Weak Comparative Baselines: Fails to compare against recent ReID-specific token pruning/selection methods (e.g., EET, DPEFormer, PAB-ReID) and other training-free pooling approaches beyond ToMe.

- Incomplete ImageNet Data: ImageNet-1k results show marginal gains but omit GFLOPs and latency metrics, making the efficiency-accuracy trade-off unclear outside of ReID.4.

- Furthermore, in the main table (table 1), the trade-off between peformance and inference seems marginal or similar, compares to ToMe. Consdiering that the ToMe method is one of the pioneer in the field of token pruning and merging, it seems that the proposed is not quite impressive. More comparsion between various pruning configuration, together with the comparison wih other recent training-free visual token pruning/merging methods in the field of ViT pruning/merging and VLM pruning/merging must be done to clarify the proposed method's practical effectiveness.

---

> ### Author Rebuttal · Authors · 2026-03-31
>
> **W1: Dataset-specific bias**, **W2: Mathematical ambiguity.**
>
> Due to space limitations, please refer to our response to Reviewer Ezgq’s W1 and Reviewer USTZ’s W3, respectively.
>
> **W3: Small value of $\delta_2$ and the importance of context.**
>
> $\delta_2$=1/20 does not flatten the context; rather, it performs a necessary numerical calibration between two distinct metrics.
> 1) **Numerical Calibration:** $F_{sim}$ is based on cosine similarity, which is typically high (≈0.8∼0.99) due to feature smoothing. $F_{res}$ is derived from the product of two normalized inverse SE-Mass, with values spanning multiple orders of magnitude (e.g., $10^{−4}$∼1) due to concentrated attention. Directly multiplying these would cause the extremely small $F_{res}$ to over-suppress feature similarity.
> 2) **Illustrative Example:** A small exponent smooths extreme numerical contrasts, e.g., $10^{−4}$ becomes ≈0.63 after $\delta_2$=1/20, and 0.0625 becomes ≈0.87. This brings $F_{res}$ into the same range as $F_{sim}$, enabling meaningful integration.
> 3) **Ablation Evidence:** As shown below (on Market-1501), $\delta_2$=0&1 lead to performance degradation, indicating that finely injecting context is crucial.
>
> |$\delta_2$|R1|mAP|
> |-|-|-|
> |0|95.3|88.9|
> |1|80.4|69.1|
> |1/20|95.8|89.2|
>
> **W4: Unanalyzed interactions.**
>
> The max-norm restoration is a *variance-preserving constraint* designed to prevent distribution collapse rather than introduce harmful shifts.
> 1) **Preventing Energy Collapse:** Standard averaging leads to energy dissipation, where feature norms contract significantly towards the origin (see Strategy1 in Fig. 7). This contraction dampens the saliency of discriminative signals before they reach downstream layers.
> 2) **Synergy with LayerNorm:** LayerNorm cannot recover the relative energy (saliency) among different tokens in a sequence. EPW-Fuse ensures that merged tokens retain a magnitude comparable to the original high-energy features before entering LayerNorm.
> 3) **Distribution Stability:** As shown in Fig. 7, the distribution of Strategy2 (ours) is significantly closer to the Original features than Strategy1. This confirms that we are *maintaining* the pre-trained feature distribution rather than distorting it.
>
> **W5: Justification for DVC and grouping strategy.**
> 1) **Rationale:** DVC maps tokens to normalized coordinates $(x_i,y_i)\in(0,1)^2$, ensuring that relative spatial positions and neighborhood relations are explicitly preserved despite changing token counts. Our alternating grouping based on SE-Mass sorting assigns tokens with similar importance (often sharing similar semantic/spatial context) to opposite groups, prioritizing the matching and merging of spatially and semantically redundant pairs.
> 2) **Visual Proof:** Fig. 8 provides direct evidence of locality preservation: background tokens aggregate within their local regions, while the spatial locality of foreground details (e.g., face) is preserved throughout the 10 layers of merging.
>
> **W6: Weak comparative baselines.**
> 1) **Comparison with ReID Models:**
> - EET: EET is not a ReID method; it is not open-source and lacks ReID benchmark results, making a fair comparison infeasible.
> - PAB-ReID: It is not open-source and performs no compression. We achieve comparable accuracy (only 0.3% Rank-1 difference on Market-1501) while saving 24% GFLOPs.
> - DPEFormer: As shown below, we outperform DPEFormer on all datasets. DPEFormer is specifically designed for occluded scenarios while we are not; nevertheless, on Occluded-Duke, our mAP is 59.4 (vs. DPEFormer 58.9), proving the robustness.
> | |Market|Duke|
> |-|-|-|
> |DPEFormer|95.4/88.1|90.0/80.3|
> |Ours|**95.8/89.2**|**90.6**/**82.0**|
> 2) **Regarding Generic Compression Methods:**
>
> Beyond ToMe , we have included comparisons with mainstream schemes like EVIT, ATS in Table 1. The results confirm that SRE-Merge significantly outperforms these generic methods.
>
> **W7: Efficiency-accuracy trade-off on ImageNet.**
>
> As shown below (use DeiT-S on ImageNet-1k), at the same merging rate (r=13), SRE-Merge shares the same GFLOPs as ToMe but achieves a 1.21% gain in Rank-1. Compared to other methods, our method shows greater advantages. This proves that our method also achieves an excellent accuracy-efficiency balance on ImageNet.
> | |R1|GFLOPs|
> |-|-|-|
> |DPC-KNN|76.89|4.02|
> |DynamicViT|77.96|5.85|
> |EViT|78.13|4.38|
> |ATS|74.97|4.14|
> |ToMe|78.28|2.83|
> |Ours|79.49|2.83|
>
> **W8: Effectiveness and scope of comparison.**
> 1) **Diverse Configurations:** Detailed comparisons under various $r$ are already provided in Table 2 and Fig. 6, where SRE-Merge consistently outperforms ToMe.
> 2) **Regarding VLM/MLLM:** ReID targets resource-constrained environments like edge devices. We utilize the ViT-B (86M), whereas massive VLMs/MLLM (even the small-scale TinyLLaVA has 3.1B) are impractical for such deployments. Due to time constraints, VLM experiments were not conducted, but our results on ViTs sufficiently clarify the practical effectiveness of SRE-Merge.

---

> > ### Author Rebuttal · Reviewer_P5do · 2026-04-05
> >
> > I appreciate the authors' efforts in the rebuttal, but I will maintain my score. While the additional results for DPEFormer and ImageNet are noted, the comparative baselines remain largely outdated and fail to reflect the current state-of-the-art in vision token pruning and merging. Specifically, the authors’ response regarding VLMs misses the point; the concern is not about the deployment of massive models, but rather the lack of comparison with recent, sophisticated pruning techniques developed for Vision Transformers within that field (e.g., those used for efficient vision encoders). Without benchmarking against these contemporary methods, it is difficult to assess whether the proposed framework offers a meaningful advancement over existing literature, especially as the performance gain over the aging ToMe baseline remains marginal despite the added algorithmic complexity.

---

> > > ### Author Response · Authors · 2026-04-05
> > >
> > > Dear Reviewer P5do,
> > >
> > > We would like to express our sincerest gratitude for your constructive follow-up feedback and for allowing us the time to conduct these additional experiments. Your insightful comments regarding the need for benchmarking against contemporary, sophisticated token reduction techniques have been instrumental in improving the rigor and impact of our work.
> > >
> > > We fully appreciate your point that the core of the evaluation should be whether our framework offers a meaningful advancement over the current state-of-the-art (SOTA) in vision token pruning and merging, including methods developed for high-performance vision encoders in VLMs.
> > >
> > > ---
> > > ### **1. Comparison with Latest SOTA Methods**
> > > Following your suggestion, we have performed an extensive benchmarking against **five representative training-free methods** published in the most recent top-tier conferences, as shown below (Rank-1/mAP). These methods represent the latest advancements in sophisticated token pruning/merging.
> > >
> > > |Methods|Venue|FLOPs (G)|Params (M)|Market1501 (%)|MSMT17 (%)|Occluded-Duke (%)|
> > > |-|-|-|-|-|-|-|
> > > |VisionZip[1]|CVPR2025|18.75|99.83|93.0/81.8|82.2/60.6|58.9/44.9|
> > > |DivPrune[2]|CVPR2025|17.84|**92.74**|94.5/88.5|84.8/67.2|66.2/58.4|
> > > |VisionTrim[3]|ICLR2026|18.79|99.83|93.0/81.2|81.7/60.6|52.4/39.8|
> > > |ApET[4]|CVPR2026|18.79|99.83|94.3/86.4|83.4/62.7|60.3/51.1|
> > > |OTPrune[5]|CVPR2026|18.79|99.83|93.0/81.0|81.9/60.4|56.3/42.7|
> > > |**SRE-Merge (Ours)**| |**15.53**|**92.74**|**95.8/89.2**|**85.3/67.8**|**67.5/59.4**|
> > >
> > > [1]VisionZip: Longer is Better but Not Necessary in Vision Language Models. CVPR2025.
> > >
> > > [2]DivPrune: Diversity-based Visual Token Pruning for Large Multimodal Models. CVPR2025.
> > >
> > > [3]VisionTrim: Unified Vision Token Compression for Training-Free MLLM Acceleration. ICLR2026.
> > >
> > > [4]ApET: Approximation-Error Guided Token Compression for Efficient VLMs. CVPR2026.
> > >
> > > [5]OTPrune: Distribution-Aligned Visual Token Pruning via Optimal Transport. CVPR2026.
> > > ### **2. Experimental Setup for Fair Comparison**
> > > To ensure a rigorous and fair evaluation, we strictly followed the protocol below:
> > > - **Implementation:** We implemented these comparison methods by directly porting their core algorithmic components from their official repositories into the TransReID backbone.
> > > - **Alignment of Constraints:** All methods were evaluated under training-free settings. We configured each baseline to achieve an identical token reduction ratio of ~38% (*SRE-Merge sets $r$=8, achieving a compression ratio of ~38%. Therefore, We align the other methods accordingly*) to directly compare their accuracy and overhead at the same compression level.
> > > - **Backbone:** All experiments use the standard ViT-B backbone to ensure that performance gains stem solely from the token reduction framework.
> > >
> > > ### **3. Analysis of Results**
> > >
> > > The results clearly demonstrate that **SRE-Merge significantly outperforms these methods in both accuracy and efficiency**. We analyze the reasons why these methods, despite their sophistication, are less effective than SRE-Merge in the context of ReID as follows: (1) Most existing methods, such as VisionZip and OTPrune, prioritize global semantic dominance or global distribution alignment, which often leads to the over-pruning or smoothing of subtle and local details (e.g., clothing logos) that are crucial for identity discrimination in the fine-grained retrieval task. (2) Unlike these methods, SRE-Merge better preserves the discriminative structural relationships between tokens by injecting spatial regularization into the merging process, ensuring that identity-sensitive information is effectively preserved rather than being discarded as redundancy.
> > >
> > > ---
> > > ### **Conclusion**
> > > By benchmarking against the most recent literature, we believe this comparison clarifies that SRE-Merge is not merely a marginal improvement over "aging baselines", but a **substantial advancement** that stands up to the latest sophisticated pruning/merging techniques in the field.
> > >
> > > Furthermore, regarding the specific accuracy-efficiency trade-off comparison with ToMe, we kindly refer you to our **Reply Rebuttal Comment** to **Reviewer ojmU** (*Part 2: Addressing Concerns on Competitive Edge Over ToMe*). We have already fully resolved the identical concerns for Reviewer ojmU in that discussion, gaining their complete endorsement of our method's effectiveness and a subsequent score increase. We believe those detailed comparisons, combined with the new SOTA results provided here, further demonstrate the practical superiority and rigorous validation of SRE-Merge.
> > >
> > > We sincerely hope that this response provides the clarity you were looking for. We would be profoundly grateful if you would reconsider your evaluation of our work in light of these significant updates.
> > >
> > > Thank you again for your time and guidance!
> > >
> > > Best regards,
> > >
> > > The authors

---

### Decision · Program_Chairs · 2026-04-30

**Decision:**

Accept (regular)

**Comment:**

The paper received WR/WA/WA/WA ratings. Reviewers were mostly concerned about the experiments (comparison with the state of art baselines, ablation study, etc). The authors provided a rebuttal. Two reviewers raised their score after rebuttal, while R1 still has concerns regarding the comparison with state of the art. AC read the paper, rebuttal and reviews, and believe the method is a good investigation on the topic and recommend accepting the paper. Authors please do an intensive literature review and comparison, comparing to the most advanced baselines for the final version.